# A Method Proposal to Adapt Urban Open-Built and Green Spaces to Climate Change

Carmela Gargiulo  and Floriana Zucaro *

Department of Civil, Building and Environmental Engineering, University of Naples Federico II,
80138 Napoli, Italy; gargiulo@unina.it
* Correspondence: floriana.zucaro@unina.it

**Abstract:** To rapidly adapt cities to the growing impacts of climate change, the open space system can play important functions as climate regulators and accelerators of sustainable urban development. To this end, this paper aims to provide a methodology that classifies open spaces on the basis of their physical characteristics and their contribution to climate vulnerability and articulates them according to the costs required for adaptation and the benefits brought. The method was applied to the city of Naples, which is an interesting case study due to its heterogeneous territory in terms of geomorphological features, such as hilly conformation and coastal location, and urban assets characterised by densely built urban fabrics with different distributions and kinds of activities. The results showed that (i) the open spaces with both low thermal and hydraulic performance are predominantly located in the peripheral part of the city, and (ii) the central area is strongly characterised by this dual issue. The latter output confirms the need to update the transformation rules of high historical-architectural value areas by introducing new resilience requirements criteria that cities are asked to have.

**Keywords:** urban open spaces; climate vulnerability; mitigation and adaptation interventions; nature-based solutions; urban resilience; Naples (Italy)

---

## 1. Introduction and Aim of the Work

Since the early 1990s, urban and territorial transformation policies and strategies have been oriented to guarantee the sustainable use of natural resources by balancing development needs that are intrinsic in complex and dynamic systems like urban ones [1,2]. The overall aim of "living well within the limits of our planet" [3] continues to be a foreground in national political agendas to remark the sustainable development concept and, then as now, it cannot be reached without rethinking methods and tools to govern cities and changing lifestyles. For instance, European households use, on average, nearly three times the amount of water than the minimum required for basic human needs (144 litres vs. 50 litres per person per day) [4]. The urban land take is still an ongoing process, with the total urban area expanding by approximately 7.3% between 2000 and 2020 [5]. The urgent need for collective action in response to these challenges is reflected in the 2030 Agenda and the Sustainable Development Goals (SDGs) that explicitly call for "transformative goals . . . transformative vision . . . [and] structural transformation". Not surprisingly, over half of the 234 indicators of the SDGs framework refer to the urban dimension so that the Recovery Plan for Europe—NextGeneration EU implementation can also be monitored [6,7].

The broad topic of how urban areas can be transformed towards the objectives of sustainable development has generated a stream of literature engaging with "urban transformation". This literature stream is oriented to study how urban areas can be rapidly and substantially transformed to become more sustainable and equitable [8,9] and, due to the urgency to face climate change impacts, to make cities inclusive, safe and climate

---

neutral. To help frame these urban transitions, policymakers are boosting a "greening transformation" of the built environment that include interventions such as street trees, parks and green open spaces, green roofs, and facades. These urban greening solutions are recognised from the IPCC special report on global warming of 1.5 °C [10] as the most suitable options for climate change mitigation and adaptation at the local scale. Furthermore, according to World Health Organization (WHO) and numerous scientific works, an interconnected network of green spaces, both on a neighbourhood and city level, provides a wide array of benefits related to energy saving, air and noise pollution reduction, managing stormwater, the extent of possible flooding reduction, urban quality, and liveability and can drive protection and enhancement of unbuilt and abandoned natural areas into spatial planning [11,12]. Reaching these positive effects requires reorganising the urban system and optimising its physical resources use to overcome the shortage of available space for the realisation of new green areas. In densely built and stratified cities such as those in the Mediterranean, it becomes strategic to transform the open space system made of existing green areas and open built-up spaces (such as squares) in a resilient way, to strengthen their eco-system capacities and to improve their adaptive capacity to climate change. This approach is also being called by the EU, which considers built and unbuilt open spaces as elements of a single system for cooling down cities and enabling long-term adaptation thinking based on bringing nature back to cities [13,14].

The key role of open and green spaces was further emphasised during the COVID-19 lockdowns, as they were recognised as essential places in urban areas to promote human health and wellbeing (see, for example, [15–18]). This increased recognition of the numerous benefits of urban greening has accompanied the launch of ambitious tree-planting programmes in many cities. Notable examples include Beijing's 50 Million Trees Programme [19], as well as the Million Trees Projects in New York, Los Angeles [20] and Singapore [21]. Cities like Vancouver, Milan and Philadelphia have become green boosters, using pro-environmental branding strategies and practices to make them more attractive and desirable places where investing and living [22]. A green city brand can be related to a vision for increased urban environmental political oversight and/or ambition to develop urban environmental qualities to gain a competitive advantage.

The restoration, enhancement and maintenance of existing urban green elements and the development of an integrated green and open spaces network provide a valuable asset to which the definition of Nature-Based Solutions (NBS) to address the local impacts of climate change can be added. NBS refer to the "concept of nature-based solutions embodies new ways to approach socio-ecological adaptation and resilience, with equal reliance upon social, environmental and economic domains" [11] (p. 15). The implementation of NBS can be particularly effective, as they include both mitigation and adaptation actions and interventions in line with the European Climate Law (EU Regulation 2021/1119) and the Recovery Plan for Europe—Next Generation EU [6], which are the main pillars for the implementation of the ecological transition in Europe.

NBS also include further green interventions like rain gardens, bioswales, retention ponds and permeable pavements useful to restore water balance by capturing, retaining and improving the infiltration capacity to adapt urban areas to flash floods and also to alleviate water stress in cities with low rainfall and/or high population density, as rainwater/stormwater is recognised as a secondary source of water [23,24]. These solutions also contribute to evapotranspiration processes and heat island effect mitigation [25].

In general, it can be asserted that within the scientific debate, it seems that alongside the numerous studies aimed at measuring the positive benefits generated by the presence of green spaces in urban areas, a segment of studies is emerging concerning the utility of spatial optimization of NBS, to help decision makers to better meet multiple sustainability objectives when developing long term urban development strategies. These studies are mainly based on an optimisation algorithm using probability-based acceptance criteria to intelligently search iteratively for better solutions. For instance, [26] applied it in a Northeast England town to demonstrate how spatial Pareto-optimisation can be employed

to derive spatial development patterns that are sensitive to climate-induced hazards, such as heatwave and flood risk. Huang et al. [27] proposed a space optimisation strategy to improve the quality and accessibility of green spaces by using this optimisation method within the urban planning process. Similarly, Zhang et al. [28] developed a multi-objective model to define the best locations and configurations for new green spaces according to their cooling effect. Multi-objective models were also used by [29,30] to improve the connectivity of green spaces and allow people to reach them by walking within a 5-min threshold. Other studies integrated fuzzy set and AHP approaches with the GIS for the assessment of land use suitability for urban green land development [31,32].

According to this scientific framework, this paper is geared toward answering the following research question: how to transform the open space system (including open-built spaces like squares and unbuilt spaces like green areas) to adapt it to climate change and increase urban resilience?

To this end, a methodology based on the following steps is provided:

- Definition of the physical characteristics of open spaces and of the urban context where they are located;
- Classification of open spaces and the neighbourhoods hosting them, according to the contribution they can provide to reduce climate vulnerability;
- Articulation of open spaces on the basis of the costs required to adapt them in terms of climate resilience and the benefits in terms of the inhabitants involved;
- Early definition of a decision support tool aimed at adapting open spaces to climate change impacts.

In other words, the main aim of the work contained in these pages is to define a methodology that reduces, on the one hand, features that may contribute to accentuating vulnerability through mitigation interventions and implements and, on the other hand, features that play in favour of resilience through adaptation interventions. The outputs of the method support local decision-makers in the definition of the most suitable adaptation interventions for open space systems, according to their physical and urban context characteristics, as well as the needed costs and the likely positive effects. The paper is structured in three parts: the first describes the methodology aimed at transforming the open space system to increase urban resilience; the second describes the results by highlighting the climate performance of open spaces; the third describes the results by highlighting the interventions to be implemented and the possible choices due to costs.

## 2. Methodology to Classify Open Spaces and Urban Areas Oriented to Climate Change Adaptation

The paper is oriented toward providing a methodology that classifies open spaces based on their physical characteristics and their contribution to climate vulnerability and articulates them according to the costs required for adaptation and the benefits brought. The methodology consists of five main phases (Figure 1).

The first step involves defining the set of variables useful for measuring the physical and performance characteristics of the open space system (squares, green areas...) for reducing climate vulnerability. In this first phase of work, we focused mainly on two types of extreme events: flooding and urban heat island (UHI). The most recent reports [10,32], which develop medium and long-term climate risk forecast scenarios at national and local levels, make it possible to identify them as the main impacts affecting different urban areas.

Heatwaves, heavy precipitation, flooding and droughts are identified as extreme climate events whose frequency and magnitude are expected to increase in Europe. South-European cities are required to face the highest projected increase in the frequency of heat waves combined with the lowest provision of green space and the most pronounced urban heat island (UHI) effect. On the other side, the increase of surface sealing in cities and the inadequacy of sewerage infrastructure to the heavy precipitation events make Mediterranean cities vulnerable to urban flooding.

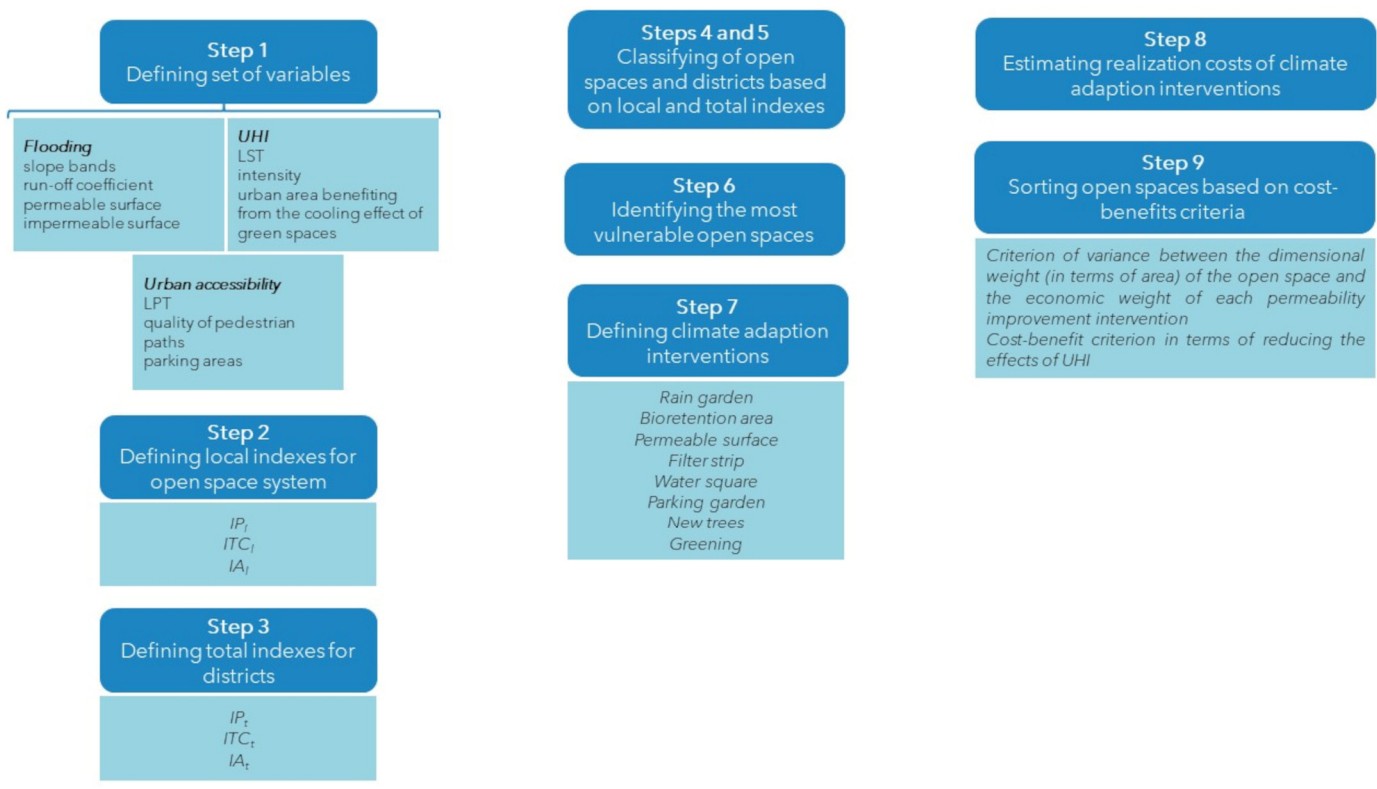

**Figure 1.** Flowchart of the proposed methodology.

For the first type of event, flooding, variables relating to morphology (slope bands), runoff capacity and surface drainage of the soil (runoff coefficient, permeable surface and impermeable surface) were identified.

These variables allow, in practice, the consideration of vulnerability to flooding as determined by the number of impermeable surfaces and the retention capacity of conventional drainage systems. As described by [33], runoff coefficients represent the percentage of runoff resulting from a storm event and are strongly influenced by land cover and soil permeability, but also by the slope and intensity of rainfall. The presence of an impermeable surface prevents precipitation from penetrating the soil, increasing the amount of runoff [34]. Major floods are more likely to occur in locations with steeper slopes than in areas with a lower slope [35].

For the second type of event, UHI, the variables refer to: the land surface temperature (LST), or "the radiant temperature measured at the interface between the surface of a material (tree canopy, water, soil, ice or snow) and the atmosphere" [36], also considered as the "skin temperature of the soil", influenced by solar reflectance, thermal emissivity and heat capacity [37]; the intensity of the urban heat island; the urban area benefiting from the cooling effect due to the presence of green spaces of a particular size [38–41].

Through evapotranspiration, solar radiation energy absorbed by leaves is converted into latent rather than sensible heat flux, thus lowering the temperature of the canopy and surrounding air [42,43]. Green areas, particularly those characterised by the presence of trees, can also lower air temperature by intercepting solar radiation, thus preventing the underlying surface from absorbing shortwave radiation, a process known as the shadow effect [44]. Based on these processes, the air temperature in urban green spaces can be 1–3 °C to 5–7 °C cooler than in neighbouring built-up areas, and this effect can also extend to the surroundings [45,46].

Such mitigation is particularly important in hot climates, such as the study area (Naples), where the heat island can elevate urban temperatures by up to 6 °C [47], posing a significant threat to the most vulnerable segments of the population to heat waves,

including the elderly, children, as well as the poorest who often do not have the possibility of benefiting from summer air conditioning systems in their residences [48].

The size of the urban area that benefits from the cooling effect, thanks to the presence of green spaces, is the result of integrated microclimatic simulations in a GIS environment (described by the authors in previous works [38–41]). These simulations allowed defining the microclimatic behaviour of the three most widespread types of urban fabric characterised by different values of built density and different sizes of green spaces (ranging from 1000 sqm to 35,000 sqm), which are likely to be found in stratified contexts. After performing simulations using the ENVI-met software, green areas sized around 5000 sqm, which can lower the average temperature by 1 °C on the surrounding built area between 100–150 m from them, were found to be more effective and efficient [38–41]. In this study, the size range of the areas due to relative climate performance is defined.

Open spaces also promote social cohesion as places of aggregation and participation, and, in this perspective, the enjoyment of these urban endowments should be guaranteed to all segments of the population. Therefore, variables related to climate vulnerability are integrated with those of improving urban usability and accessibility to contribute to the improvement of citizens' quality of life (step two). The variables are related to open spaces accessibility, such as Local Public Transport (LPT) (no. of stops near the green area), quality of pedestrian paths (width of sidewalks, quality of pavement, etc.) and parking areas (no. of parking spaces).

After collecting the related data, the database was populated in a GIS environment using the open-source software QGIS (step three). Except for the eight variables related to the runoff coefficient, permeable and impermeable surfaces, land surface temperature, cooling distance, cost and population, the initial quantitative values of the other five variables (slope bands, heat island intensity and accessibility) were classified into three qualitative ranges (low, medium, high).

The measurement of the 13 variables for each open space is followed, in the fourth step, by the overall measurement of permeability, thermal comfort and accessibility to assess the current functioning/performance of the entire open space system about each of the objectives. For this purpose, the following three synthetic indices were defined:

$$IP_l = (\text{Runoff coefficient + Permeable surface + Impervious surface + Slope})/n, \quad (1)$$

$$ITC_l = (\text{LST + UHI + Cooling area})/n, \quad (2)$$

$$IA_l = (\text{Parking accessibility + LPT accessibility + Pedestrian accessibility})/n, \quad (3)$$

where $IP_l$ is the Local Permeability Index, $ITC_l$ is the Local Thermal Comfort Index, $IA_l$ is the Local Accessibility Index, and n is the number of variables defined for each objective.

To measure these indices, it was first necessary to standardise the scale of values of some variables of the $IP_l$ and $ITC_l$, due to the different "significance" that a high or low value determines for each objective and then to carry out the normalisation. For example, a high value of the permeable surface area favours the improvement of permeability, in contrast to the case of the impermeable surface area and the runoff coefficient, which contribute to the achievement of the objective when characterised by low values.

The characteristics of individual open spaces, both for improving permeability and thermal comfort, also depend on the characteristics of the urban context in which they are located. Indeed, the morphology and layout of the territory contribute to determining the "response" of the open space to an extreme climatic event, also conditioning its usability.

With this in mind, a Total Permeability Index $IP_t$ and a Total Thermal Comfort Index $ITC_t$ are calculated for each neighbourhood to assess the capacity of the territorial context to contribute to reducing the risk of flooding and heat island risk and to classify the

neighbourhoods concerning these 2 climatic aspects (taken individually) identifying the most critical ones:

$$IP_t = (\text{Runoff coefficient} + \text{Natural surface} + \text{Impervious surface} + \text{Slope})/n, \quad (4)$$

$$ITC_t = (\text{UHI} + \text{Building density})/n, \quad (5)$$

where the values of the two variables are the average values for the neighbourhood. For the calculation of this index (step five), unlike the $ITC_l$ of individual open spaces, the density of the built-up area was taken into account both because this variable is closely related to the phenomenon under consideration, as it contributes to the storage of heat in the urban area by capturing a large part of the incident solar radiation, and is therefore relevant in considering the context characteristics, and because the value of the area affected by the cooling effect determined by the presence of a green space already takes into account the density characteristics of the urban fabric in which the open space is inserted [38–41].

Steps four and five allow for the identification of neighbourhoods that constitute "warning areas" within the urban territory.

In step six, within the individual neighbourhoods, the open spaces characterised by the lowest and highest values of the local $IP_l$ and $ITC_l$ indices, respectively, are identified and thus constitute the priority ones for which appropriate adaptation solutions should be provided. The proposed intervention for each of these open spaces is to be carried out to contribute simultaneously to both the improvement of permeability and thermal comfort, thus optimising the use of the resources available to local administrators and taking into account that the reference context is the stratified historical city, typical of the European reality. The choice of intervention is such to guarantee the compatibility of the transformation with the intrinsic characteristics of the space (such as the surface, the slope and the current use) and with those of the urban context in which the space is inserted (such as the type of fabric and its historical-artistic-architectural value).

In step seven, adaptation interventions are defined with reference to the National Climate Change Adaptation Plan [49] and to the digital platforms and guidelines developed by the European Union to increase resilience through appropriate transformations (e.g., Climate-ADAPT, Blue App. Climate-ADAPT, Blue App; [50]).

The selected interventions include both mitigation interventions (such as permeable surfaces and parking gardens) to reduce the contribution of open space features to climate vulnerability and adaptation interventions (such as greening with tree species that promote microclimate regulation, filtered strips, bio-retention areas, rain gardens, water squares) aimed at strengthening features that make a positive contribution to urban resilience.

In step eight, the effectiveness of the proposed interventions is assessed by estimating their implementation costs to subsequently sort out both the open spaces and the relevant neighbourhoods where it is appropriate to intervene.

The calculation of the costs requires referring to some guide criteria for the design of the interventions [11,50,51] to have the first reliable quantification of the financial burdens that the local administration would have to bear for the implementation of the interventions (Table 1).

**Table 1.** Main criteria to estimate intervention costs to improve the permeability of open spaces.

| Adaptation Intervention for Increasing Permeability | Slope | Surface |
| --- | --- | --- |
| Rain garden | <8% | <8000 sqm |
| Bioretention area | <10% | <8000 and at least 200 sqm |
| Permeable surface | <5% | <15,000 sqm |
| Filter strip | <5% | - |
| Water square | <6% | - |
| Parking garden | <6% | - |

The hydraulic and thermal modelling of individual interventions, which are useful for detailed design purposes, can be integrated at a later stage of the work, further verifying it with respect to the more purely engineering aspects. In fact, this paper is aimed at providing an initial cognitive and methodological result for the resilient transformation of the open space system.

The cost estimate refers to the *Prezzario delle Opere Pubbliche* (Public Works Price List), which is the reference tool for the prior quantification, design and realisation of regional public works, as required by Article 23 of Legislative Decree 50/2016 (Contracts Code). Table 2 shows the main costs of the proposed adaptation interventions.

**Table 2.** Main criteria to estimate costs of interventions to improve the permeability of open spaces.

| Climate Adaptation Intervention | Estimated Unit Cost [€/sqm] | Climate Vulnerability | |
|---|---|---|---|
| | | UHI | Flooding |
| Rain garden | 70 | x | x |
| Bioretention area | 80 | | x |
| Permeable surface | 35 | | x |
| Filter strip | 150 | | x |
| Water square | 250 | | x |
| Parking garden | 110 | x | x |
| New trees * (in existing green areas) | 64 * | x | x |
| Greening | 69 | x | x |

* Cost per single tree.

In step nine, the effectiveness of adaptation interventions is assessed for each climatic problem that may characterise an open space: low permeability, high thermal stress, and coexistence of both conditions. Effectiveness is evaluated based on two criteria:

- The criterion of variance between the dimensional weight (in terms of area) of the open space and the economic weight of each permeability improvement intervention to be carried out; the second weight refers to the economic charge of the same type of intervention on all identified attention districts;
- The cost-benefit criterion in terms of reducing the effects of UHI is based on the number of inhabitants that fall within the cooling area (determined, as we have already mentioned, by the type of urban fabric and the size of the green area itself). The number of inhabitants is related to the cost of the relevant greening intervention to obtain the cost that each inhabitant would have to bear to take advantage of the thermal improvement.

Evaluating the effectiveness of adaptation interventions based on these two criteria provides local decision-makers with a sorting of the open spaces that are in priority need of adaptation to increase the resilience of the urban system. This output represents the first tool that guides the public decision-maker in choosing the most effective intervention to be implemented.

### 3. Study Area: Physical Characteristics and Urban Context of the Open Space System in the City of Naples

The Municipality of Naples (Latitude 40.8517746 and Longitude 14.2681244) is the capital of the Campania Region in Southern Italy and takes a key role in the Italian urban structure as it is the centre of a very wide metropolitan system and embraces great social, economic and cultural contradictions. The city lies over an area of 118 km$^2$ and, with around 950,000 inhabitants and a population density of approximately 7.754 inhabitants/km$^2$ [52], is among the most populated Italian cities. Naples is characterised by a progressive ageing of the population, with rates above the national average [53], and it is still among the European cities where the population aged 65 and over is expected to be higher the 25% of the total number of inhabitants [54].

According to [55,56], this kind of demographic scenario represents "a risk due to a combination of exposure and increased psychosocial susceptibility or social vulnerability as older people are more susceptible to disease and the effects of stresses on the food and water supply, and reduced ability to mobilise themselves in an extreme weather event". In this perspective, the study case of Naples can provide effective insights into the development of a comprehensive set of adaptation measures, actions, and interventions that can feed into the current development of the city resilience plan. In our case, the impacts considered about the objective of reducing climate vulnerability are heat waves and flooding due to intense rainfall, which will tend to occur with greater frequency and intensity: more than 90 consecutive days of temperatures above 37 °C and intense rainfall every 4 years instead of every 10 [56].

Based on these forecasts, objectives were identified to improve urban permeability, help facilitate the drainage of rainwater, and to improve thermal comfort, to encourage a decrease in the heat island phenomenon and the consequent energy consumption related to summer air conditioning. In this regard, [57] estimated that "an additional 235 billion euros of investment and operational expenditure will be required for the generation and transmission of electricity for space cooling" in the absence of appropriate interventions and adaptation actions.

Most of the data useful for measuring the variables were retrieved through processing in a GIS environment from open databases, such as ISTAT for population, Urban Atlas for area rates and Open Street Map for the road network. In particular, the physical and geometric characteristics of the arcs of the pedestrian network within a 400 m area of each open space were identified punctually, and the stations of the rail network and stops of the road network were geolocated to measure pedestrian and LPT accessibility, respectively. For parking allocations, data retrieved from Open Street Map were integrated with those from the Sustainable Urban Mobility Plan of the city of Naples currently being drafted [58]. The definition of the three classes for the accessibility variables was done considering the willingness of the most fragile segments of the population to walk to reach open space, for which the distance of 400 m is the maximum distance to walk due to their behaviours.

Raster image processing was carried out for runoff coefficient, slope, and temperature variables. Specifically, for the slope, it was necessary to process the digital terrain elevation model (DEM) of the study area and then carry out the acclivity analysis, which allowed defining 4 bands (low, medium, high and high slope).

The measurement of temperature and related urban heat island intensity values, on the other hand, required the processing of multispectral and thermal data from Landsat 8 satellite images, which are made available from the U.S. Geological Survey website and are among the most effective for monitoring and mapping the environment at the spatial level [59,60]. Specifically, a medium-resolution (30 m/pixel) raster image was processed to analyse the spatial variation of air temperatures at the urban layer between the ground surface and 2 m, normally referred to as the canopy layer, about vegetation (Figure S6). From the temperatures of the canopy layer for the day 25 July 2022 (the date at which an image of the study area characterised by the almost complete absence of clouds was available), the urban heat island was calculated as the difference between the average temperature measured for the urbanised area and the average temperature measured in the non-urbanised (rural) area (Figure S7).

Naples case study is also interesting due to its heterogeneous territory in terms of geomorphological features, such as hilly conformation and coastal location, and urban assets characterised by densely built urban fabrics with different distributions and kinds of activities.

The city has undergone an urban transformation process over time [61,62]: starting from the 1990s, a strong planning framework was developed to recover the largely derelict industrial area of Bagnoli (the western part of the city, Figure 1); to enhance the historical central area (e.g., Montecalvario, Avvocata and San Ferdinando districts, Figure 2) by rehabilitating residential buildings, restoring and reusing other historic buildings, and

transforming public spaces in pedestrian areas; to regenerate the Eastern periphery (Barra, Ponticelli, Secondigliano, Figure 2) where building public infrastructures and new collective functions.

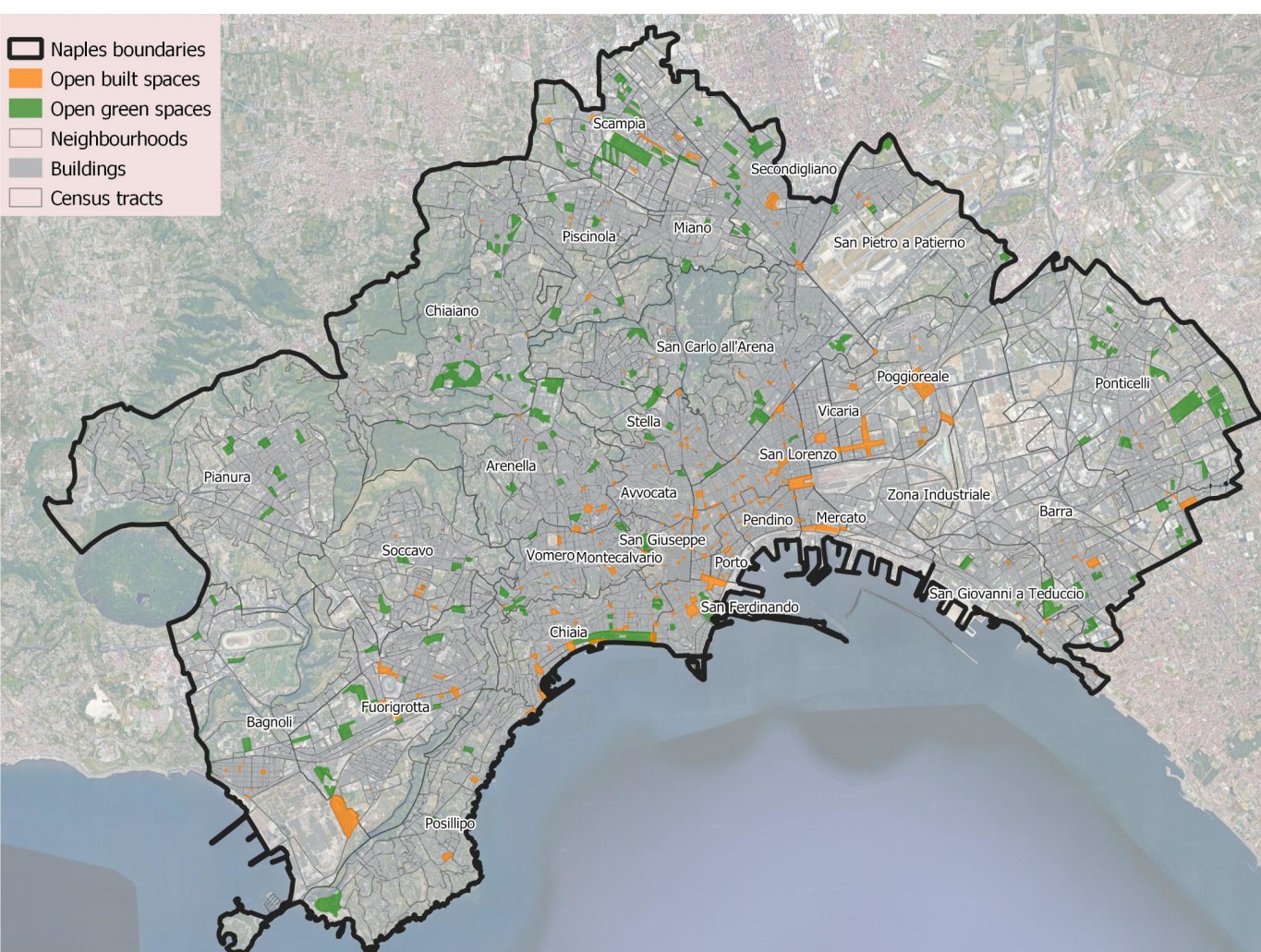

**Figure 2.** Localisation of open spaces (green and open built areas) in the 30 Naples districts.

Therefore, the open spaces system in a densely stratified built city like Naples represents a relevant resource to increasing urban resilience by cooling the built environment, improving stormwater management, and encouraging sustainable mobility.

Figure 2 shows the system of 179 green areas and 266 open-built spaces located in Naples. The first includes districts and green pocket spaces also equipped for play and sports, while the second refers to squares and sealed but unbuilt areas that are meeting and exchange places, shared places of urban living. Both kinds of spaces refer to the public ones with a minimal surface of 55 m² since, below this value, it is no longer a space but an element of street furniture (e.g., a roundabout with vegetation or a road intersection area). Zona Industriale is the only neighbourhood where there are no open spaces due to its manufacturing land use.

The distribution of the open space system is indicative of the urban planning processes, or lack of such, that have determined the urban asset of different parts of the city. In the consolidated central neighbourhoods, the result of a unified urban project, the co-presence of both types of spaces can be identified: this is the case in Vomero, Chiaia, San Ferdinando and partly Arenella. The adjacent neighbourhoods such as Avvocata, Montecalvario, San

Lorenzo, and Porto are characterised by the exclusive presence of impermeable open spaces whose dimensions and forms highlight not always controlled urban development processes.

It is worth noting that during the early 2000s, the open spaces (and main streets) of this central part of the city were interested in numerous urban redevelopment interventions aimed at favouring pedestrian usability to improve the tourist attractiveness of the relevant cultural and architectural heritage. The increase in pedestrian usability resulting from such interventions, as well as adequate accessibility through transportation offerings, significantly characterises neighbourhoods such as San Ferdinando, Chiaia, Vomero, Montecalvario, and Porto (Figures S3–S5).

The widespread presence of small-sized open spaces characterising the most stratified part of the city contrasts with those of larger dimensions located above all in the Northern suburban districts such as Scampia and Secondigliano. The lack of maintenance of these spaces located within the impressive public residential building complexes (the best-known example is Vele di Scampia) and of safety perceived by people when using them contribute to making these districts anonymous neighbourhoods with a low quality of life. The related open spaces are characterised, overall, by lower runoff coefficient values than those located in the central area of the city, due to the different urban fabric that appears to be of a unified design and recent formation.

Finally, if the limited presence of open spaces in the western districts of Barra, San Giovanni is attributable to their main productive connotation, in the eastern districts such as Pianura, Soccavo and Bagnoli, the "aggressive" building has speculated on spontaneous settlements, to the detriment of the provision of public spaces and collective services. The open space system of the eastern part of the city, as well as the western part of the city, appears to be characterised by numerous deficiencies in terms of both accessibility, especially pedestrian accessibility and adaptability to the impacts of climate change, due to the high values of UHI and runoff coefficient (Figures S1, S3 and S7).

## 4. Results and Discussion of the Classification of Open Spaces and Neighbourhoods According to Their Contribution to Reducing Climate Vulnerability

The objectives of improving permeability, thermal comfort and accessibility were measured by aggregating the respective variables into appropriate indices for each of the 445 open spaces in the study area (step 4 of the methodology, Figure 1).

Starting from Figure 3a, which shows the classification of open spaces considering local permeability index ($IP_l$) values, it can be seen that these are strongly characterised by a lack of drainage capacity. Almost 73% of the spaces are found to have low $IP_l$ values, and this may be attributable to the type of soil and/or the type and maintenance of the drainage pavement present (permeability decreases in part over time due to the accumulation of dust in the joints between the slabs). This percentage of open spaces with low IPl is widespread in most of the neighbourhoods of the city of Naples, except Pianura, Bagnoli, San Carlo all'Arena, Piscinola and Chiaiano, which are instead predominantly characterised by spaces with medium and high permeability; in particular, the last two neighbourhoods just mentioned include almost the few spaces (40) with the best water drainage performance.

The open space system of the city of Naples is characterised by an average $ITC_l$ of 47%, distributed mainly in the districts of Fuorigrotta, Scampia, Porto, Vomero, and Poggioreale. This result can be attributed, on the one hand, to the cooling effect due to the contiguity between open spaces, causing an amplification of the cooling effect (Figure S2) and, on the other hand, to the circumstance that in these neighbourhoods, the LST values do not exceed 31 °C on average (Figure S6).

It is interesting to note that open spaces with a low $ITC_l$ value characterise almost 36% of the 455 spaces, which fall almost entirely in the districts of Pianura, Arenella, Piscinola, Ponticelli and Barra, confirming the relevance and urgency of defining effective adaptation solutions in the face of both heat waves and flooding, in the light of what has been described for $IP_l$. Scampia, San Carlo all'Arena, Stella, Chiaia and Vomero are, finally,

the neighbourhoods in which most of the 73 spaces with low $ITC_l$ are located (Figure 3b) thanks to the consistent presence of green spaces (Figure S2).

As far as $IA_l$ is concerned, open spaces turn out to have, on the whole, medium-high accessibility (about 73% of the total, Figure 3c) due to the adequate supply of both the LPT and pedestrian network (Figures S3 and S4). These are, for example, the open spaces located in the neighbourhoods of Arenella, Vomero and San Ferdinando, neighbourhoods characterised by significant tourist attractiveness, and those found in Fuorigrotta, Scampia and Vicaria, neighbourhoods with an adequate pedestrian network.

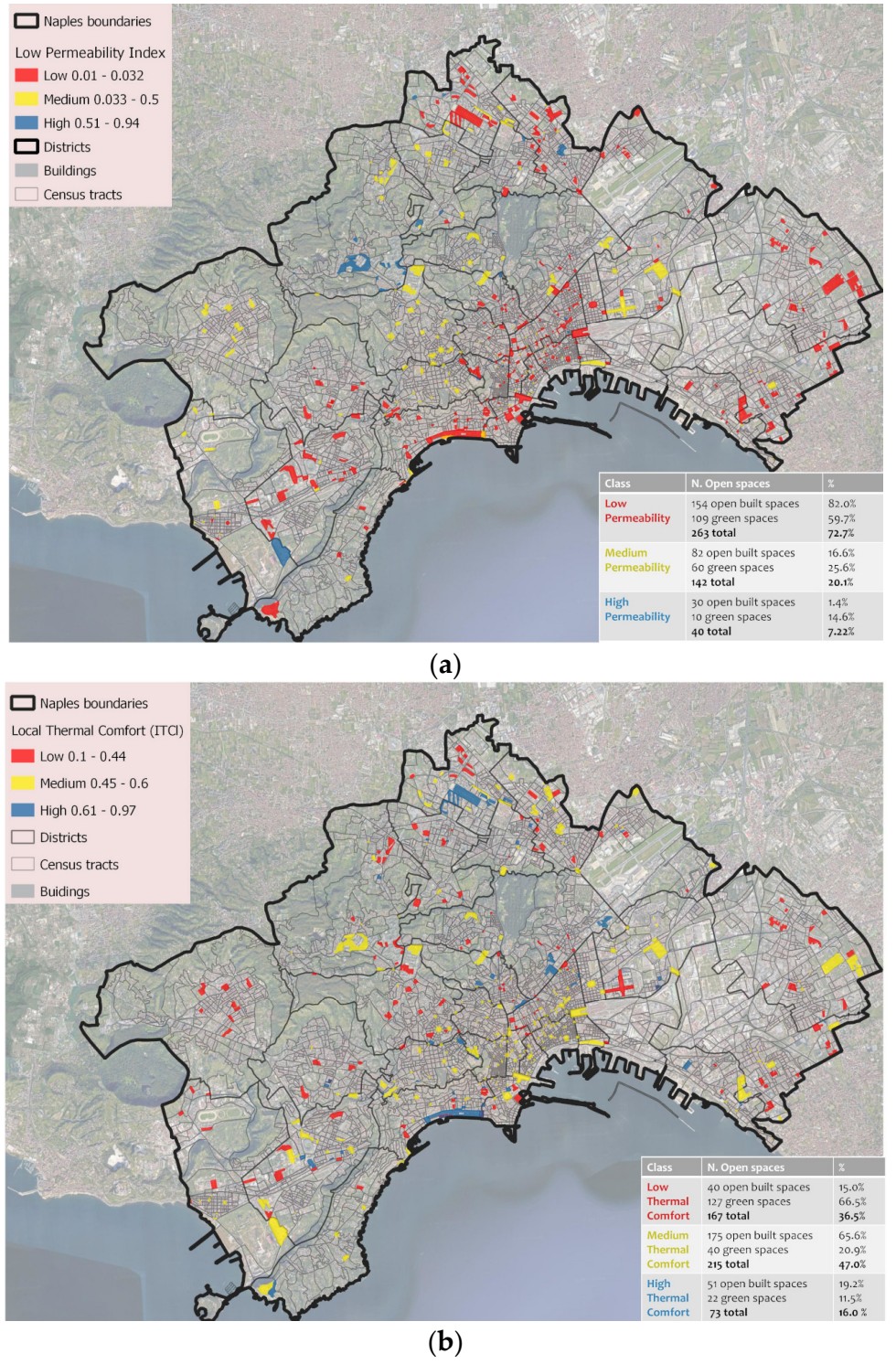

(**a**)

(**b**)

**Figure 3.** *Cont.*

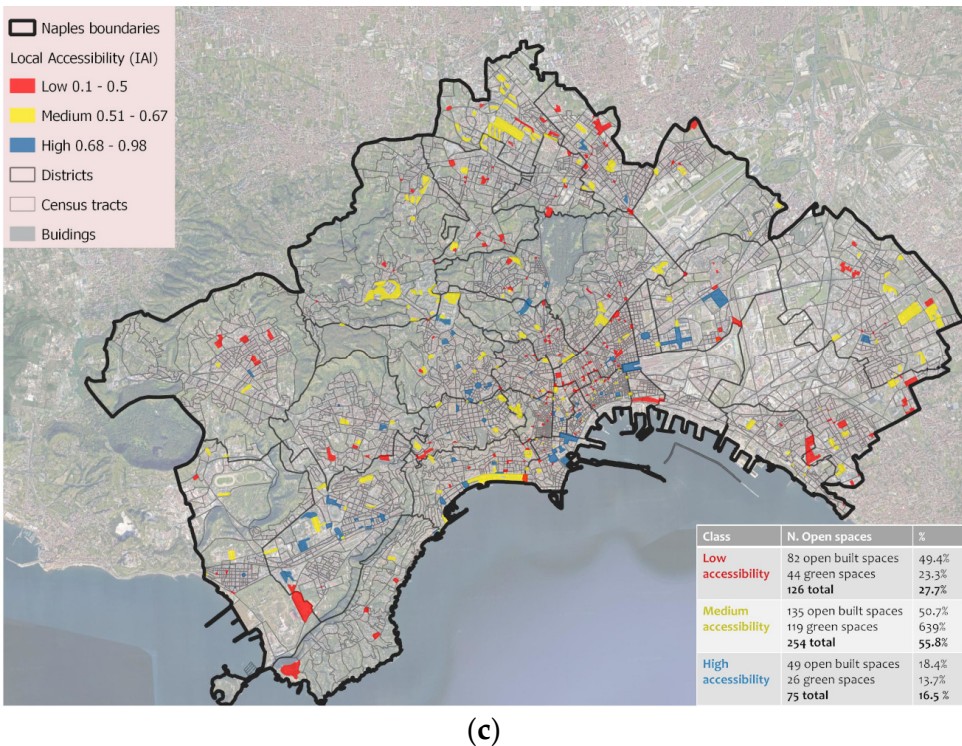

(**c**)

**Figure 3.** Classification of open spaces system according to the Index of Local Permeability (**a**), Index of Local Thermal Comfort (**b**) and Index of Local Accessibility (**c**).

The measurement of the permeability and thermal comfort indices of the open space system was followed by that of the neighbourhoods to identify the "warning areas" regarding these two aspects (step five of the methodology, Figure 1). In general, it is possible to note that neighbourhoods such as Pianura, Chiaiano, Bagnoli and San Carlo all'Arena are characterised by a significant presence of natural surface (between 2 and 5 sq km), as shown in Figure S8; furthermore, the same figure shows the $IP_t$ values (which are high) for the calculation of which the rate of unbuilt territory was considered. Within these neighbourhoods, open spaces reach medium-high $IP_l$ values, which seems to demonstrate the key influence of context factors such as permeability.

In the rest of the city, there are neighbourhoods, mainly located in the central and eastern area, with low $IP_t$ values (Figure 4a) and high permeability of individual open spaces, and neighbourhoods, mainly located in the western area, with medium $IP$ values and low permeability of individual open spaces, which can be attributed to the intense degree of sealing.

As far as the thermal comfort index on a neighbourhood scale is concerned, the urban area of Naples is characterised by medium-high values of $ICT_t$ (Figure 4b) due to the high values of both LST, which strongly characterise the municipal territory, and building density, which contributes to exacerbating the UHI phenomenon (Figures S6, S7 and S9).

This is the case of Arenella, Avvocata, San Lorenzo and Barra, with high values even of $ITC_l$ relative to open spaces (Figure 3b). In neighbourhoods such as Posillipo, Pianura, Soccavo and San Carlo all'Arena, the key role of vegetation in terms of regulating the urban microclimate is evident, which contributes to determining, on the whole, an average $ICT_t$, with the same values of built density (Figure S9).

Chiaiano, Bagnoli and San Giovanni a Teduccio are, finally, the neighbourhoods characterised by low $ICT_t$ (Figure 4b) values due to the medium-low values of both UHI and building density, but where the open spaces present a lack of thermal comfort attributable to the presence of sealed surfaces and high emissivity materials, which contribute to storing solar radiation (Figure S7).

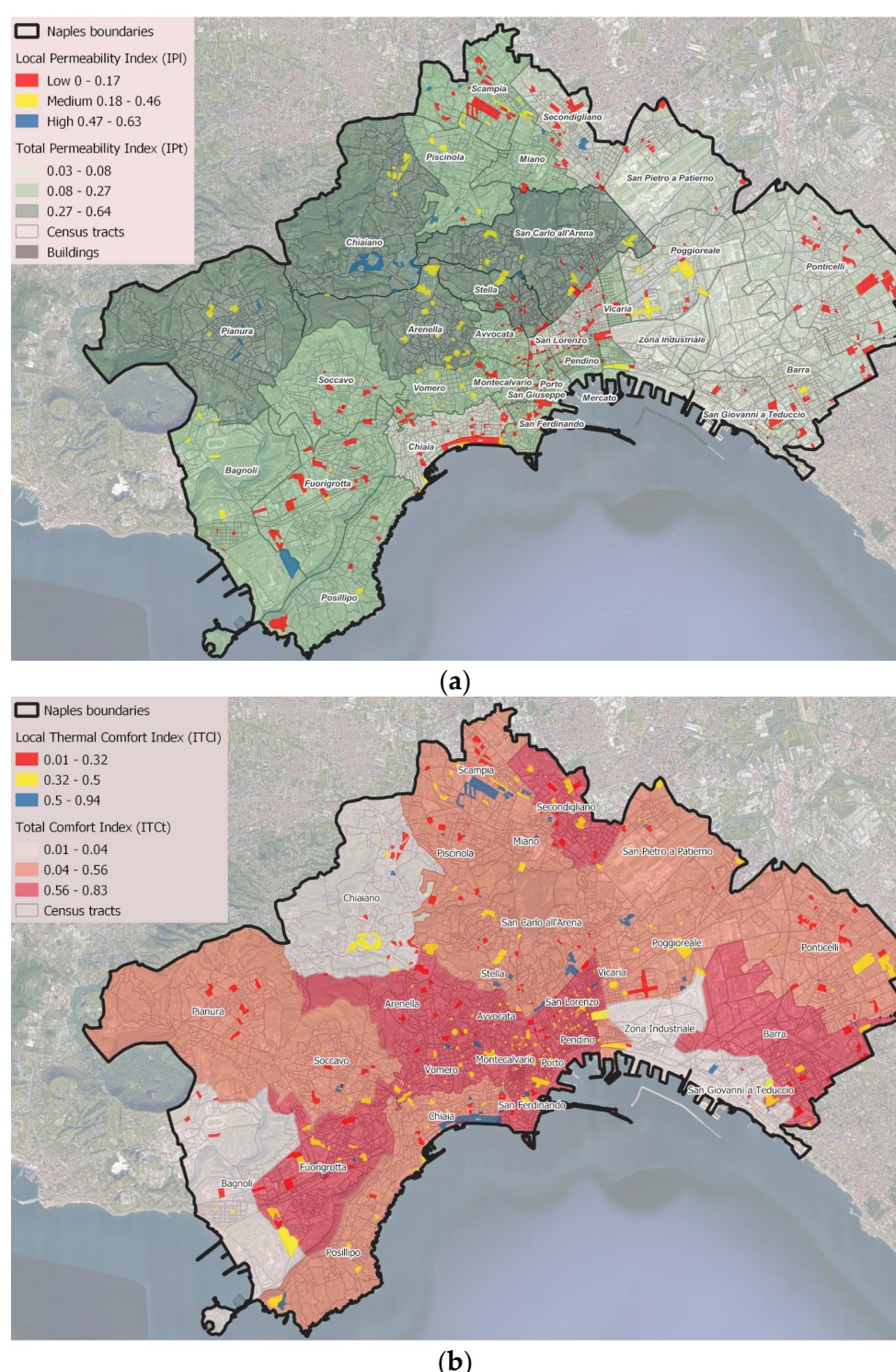

**Figure 4.** Classification of neighbourhoods and open space systems according to the total and local Permeability Index (**a**) and to the total and local Thermal Comfort Index (**b**).

The "warning areas" are the neighbourhoods that have $IP_t$ and $ICT_t$ values, respectively lower and higher than the average ones (Figures 5 and S10). The coexistence of these conditions results in a high climatic vulnerability for 11 neighbourhoods located mainly in the central and eastern areas of the city of Naples: Avvocata, Barra, Fuorigrotta, Montecalvario, Pendino, Poggioreale, Ponticelli, San Ferdinando, San Lorenzo, Secondigliano and Vomero.

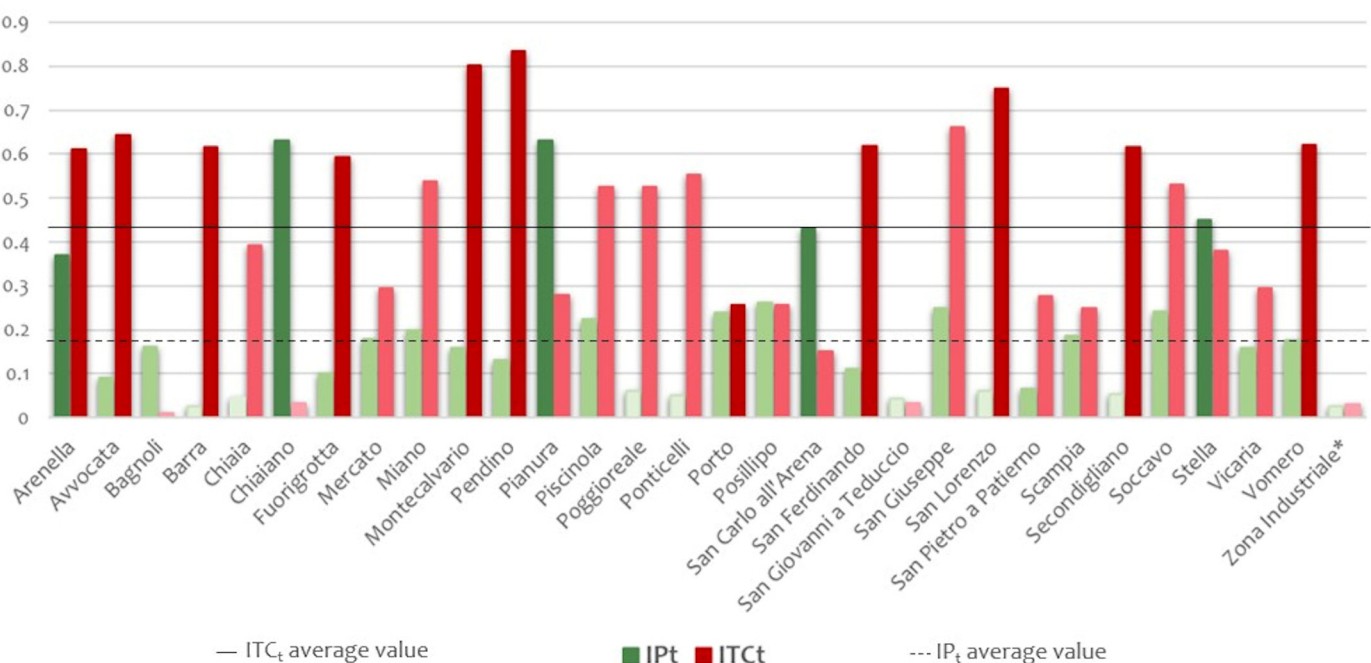

**Figure 5.** Total permeability and thermal comfort indexes of Naples neighbourhoods and related average values. * Zona Industriale has not open spaces.

## 5. Results and Discussion of the Classification of Open Spaces Based on the Costs and Benefits and Definition of the Decision Support Tool

In these warning neighbourhoods, the open spaces characterised by the worst climatic performance were taken into consideration, i.e.,: those with a low $IP_l$ value, those with a high $ICT_l$ value and those with the "critical" values of both indices at the same time. The occurrence of one of these conditions would require the implementation of an adaptation intervention, which was proposed with an estimate of the main implementation costs. Each solution was suggested by considering the main climate vulnerability, the physical characteristics of the open space and the neighbouring urban context. These three elements are oriented to guarantee that the interventions are consistent with the existing land use and urban asset to reach transformation compatibility.

First, the results of step 6 of the methodology (Figure 1) related to the different climate adaptation interventions that were proposed for the open spaces located in the "warning districts" are presented and discussed (Sections 5.1–5.3). Next, the results of steps 8 and 9 of the methodology (Figure 1) related to the sorting of the proposed interventions, according to their effectiveness assessed in terms of costs and potential benefits, are presented and discussed (Section 5.4).

### 5.1. Open Spaces with Low $IP_l$ Value

Starting with the 77 open spaces characterised by permeability problems (Figure 6a,b), these are mainly located in the historic and consolidated neighbourhoods in the central area of Naples, such as Avvocata, Pendino and San Lorenzo. While in the Avvocata neighbourhood, the open spaces are almost empty spaces enclosed in the dense built-up fabric, which extends to the slopes of the Arenella hill area, in the adjacent neighbourhoods of San Lorenzo and Pendino, the system of open spaces consists of numerous squares, some of which are the result of the redefinition of the street grid and urban fabric that took place at the end of the 19th century, such as Piazza Nicola Amore and Piazza Calenda.

This urban layout has made it possible to propose "small-scale" interventions (rain gardens, filtered strips, bioretention areas) to improve the permeability of the system of open spaces located in these neighbourhoods of a historical layout, as well as those in the Vomero district. For the open spaces where there are areas for parking, the proposed

intervention is the parking garden to ensure functional compatibility with a view to greater sustainability, especially in the central area (Pendino, San Lorenzo), where finding new spaces for parking would not be an easily achievable objective.

It is worth noting that for two open spaces located in the neighbourhoods of Secondigliano and Vomero, integrated interventions have been proposed (filter strips and rain garden in the first case, bioretention and filter strip in the second case) to improve the permeability of the unbuilt surface area and facilitate drainage also by improving the channelling of rainwater, due to the limited surface area available.

The greater extension of open spaces in the Poggioreale, Secondigliano and Ponticelli neighbourhoods also is appropriate for interventions such as water squares. The latter allows to satisfy both the needs for temporary water storage during heavy rainfall as well as those for the redevelopment of public spaces, key places for aggregation and participation in neighbourhoods characterised by phenomena of social distress such as those of the north-eastern suburbs of the city of Naples.

Figure S11 shows that interventions aimed at improving permeability alone amount to approximately 26 million euros, with larger investments in the Vomero, Fuorigrotta and Poggioreale neighbourhoods due to the larger and more numerous areas in which to intervene. Almost all of the open spaces also fall within the historic centre, which is also recognised as a UNESCO heritage site, which implies the presence of urban planning rules and regulations oriented towards protecting the heritage of historical, architectural and cultural interest to the detriment of possible transformations that climate change has now made unavoidable.

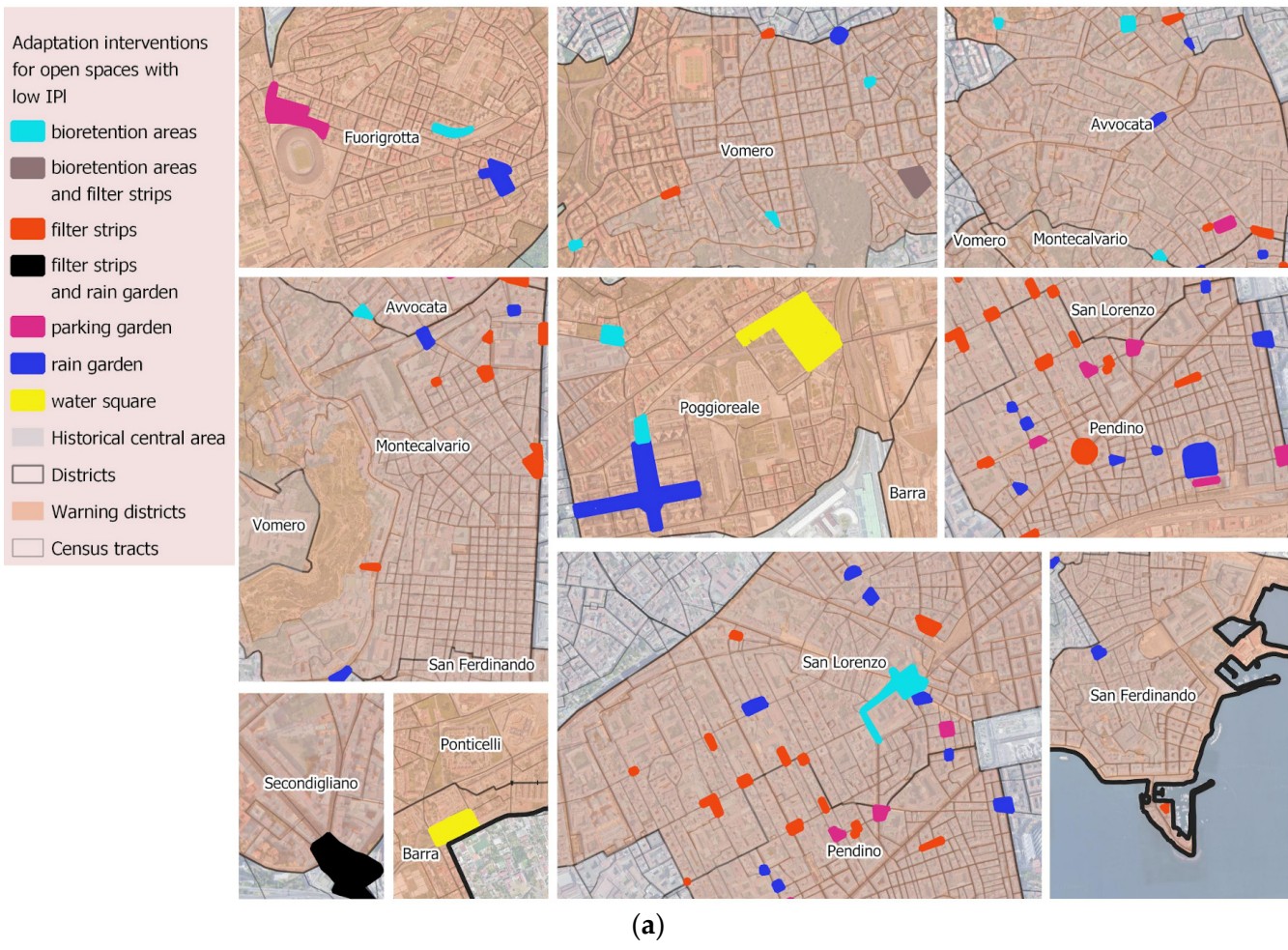

(**a**)

**Figure 6.** *Cont.*

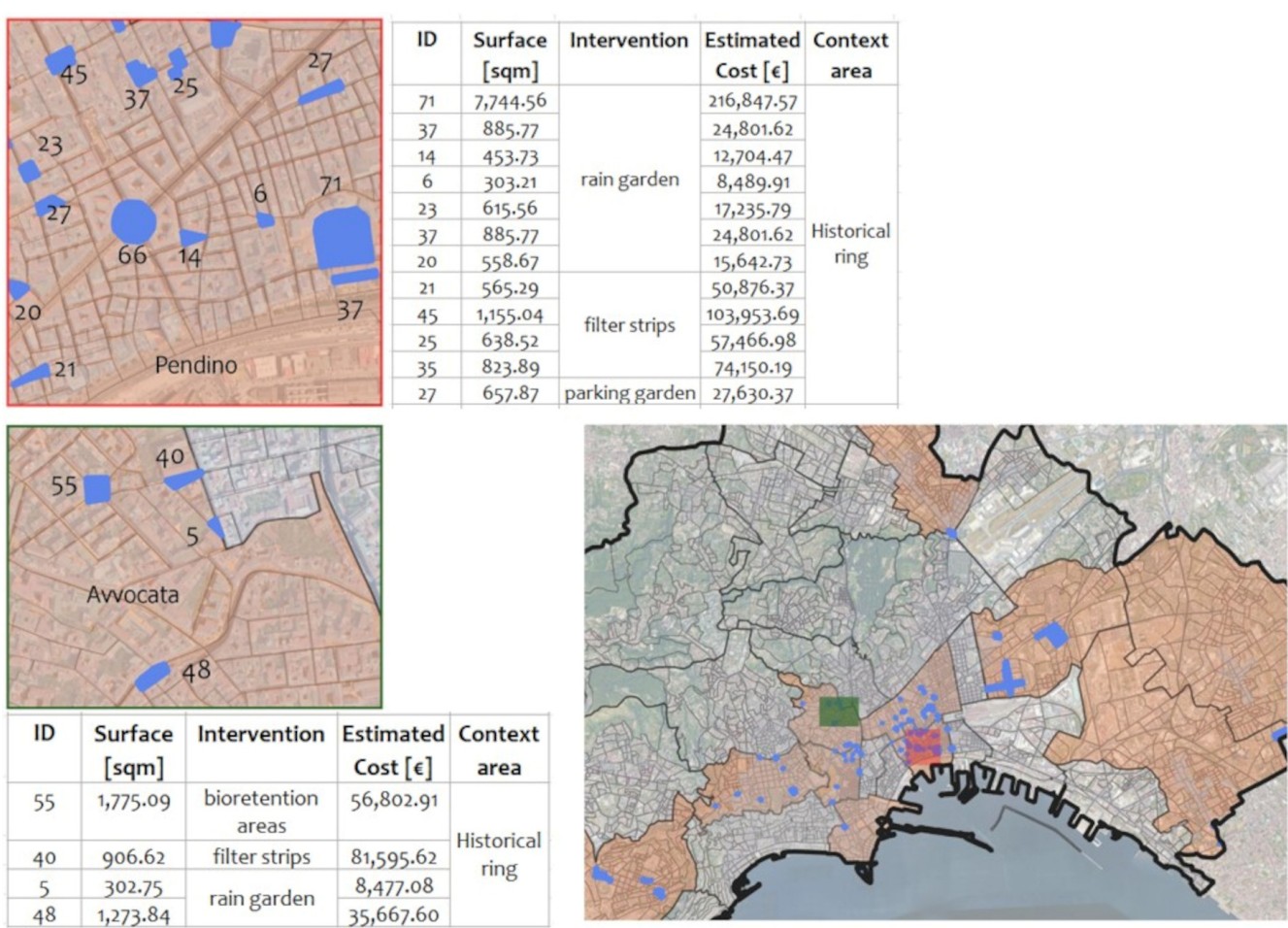

**Figure 6.** Classification of open spaces with local Low Permeability Index in warning districts (**a**) and some examples of interventions and related costs of open spaces with a Low Permeability Index (**b**).

### 5.2. Open Spaces with Low $ITC_1$ Value

The 31 open spaces found to be deficient in terms of thermal comfort are mainly located in the central-western part of the city (Figure 7a,b). Here, the neighbourhoods of Fuorigrotta and Vomero are mostly green areas where, presumably, the evapotranspiration process is affected by the UHI phenomenon and the intense presence of built-up areas, especially those located in the latter neighbourhood (Figures S7 and S9). In the Secondigliano neighbourhood, on the other hand, the open spaces are areas intended for parking, except a larger area characterised by the presence of an extensive green area. In the latter case, the proposed intervention is aimed at increasing the number of trees to help improve the cooling effect; this solution also concerns the other green spaces located in Fuorigrotta and Vomero.

For the remaining open spaces not currently characterised by the presence of vegetation and distributed both in the districts just mentioned and in the remaining ones in Montecalvario, San Ferdinando, San Lorenzo and Barra, the suggested interventions are those of greening to mitigate the effects of the UHI and contribute to the reduction of energy consumption.

The costs to be borne for the implementation of these types of interventions amount to just under 5 million euros, with the highest rate due to the ex novo planting of tree species which also concerns areas located within the historical centre of the city of Naples (Figure S12).

### 5.3. Open Spaces with Low IP_l and ITC_l Values

Turning finally to the 58 open spaces characterised by both thermal comfort and permeability problems, these are distributed in almost all the neighbourhoods that constitute the city warning areas identified above, with the exception of Pendino and Poggioreale (Figure 8a,b). It is worth noting that the neighbourhoods of Fuorigrotta, Secondigliano, Barra and Ponticelli are almost exclusively green areas subject to significant thermal and stormwater runoff loads caused by the highly impermeable context in which they are located. This state of affairs may be ascribable to a process of both expansions that have not always been planned and orderly and redevelopment that does not yet seem to have fully valorised and renewed urban places, also because of the current climatic-energetic scenarios. The only exception in this respect is an open space intended for parking located near the Maradona stadium in the Fuorigrotta district.

Moving to the central area of the city, numerous open spaces are located in the San Ferdinando district and play an important role in the usability and attractiveness of this area, given their proximity to buildings and places of cultural and architectural interest, as well as their intrinsic historical value. This is the case of piazza Municipio, the Molosiglio area, and piazza Santa Maria degli Angeli, to name but a few. To these can be added piazza Montecalvario, located in the district of the same name, which constitutes one of the few voids within the stratified building fabric, and piazza Giannone and piazza Carlo III in the San Lorenzo district which, although close to each other, differ in size (the former has a limited surface area compared to the latter, which is among the city's largest squares) and in current use (the former is entirely intended for parking).

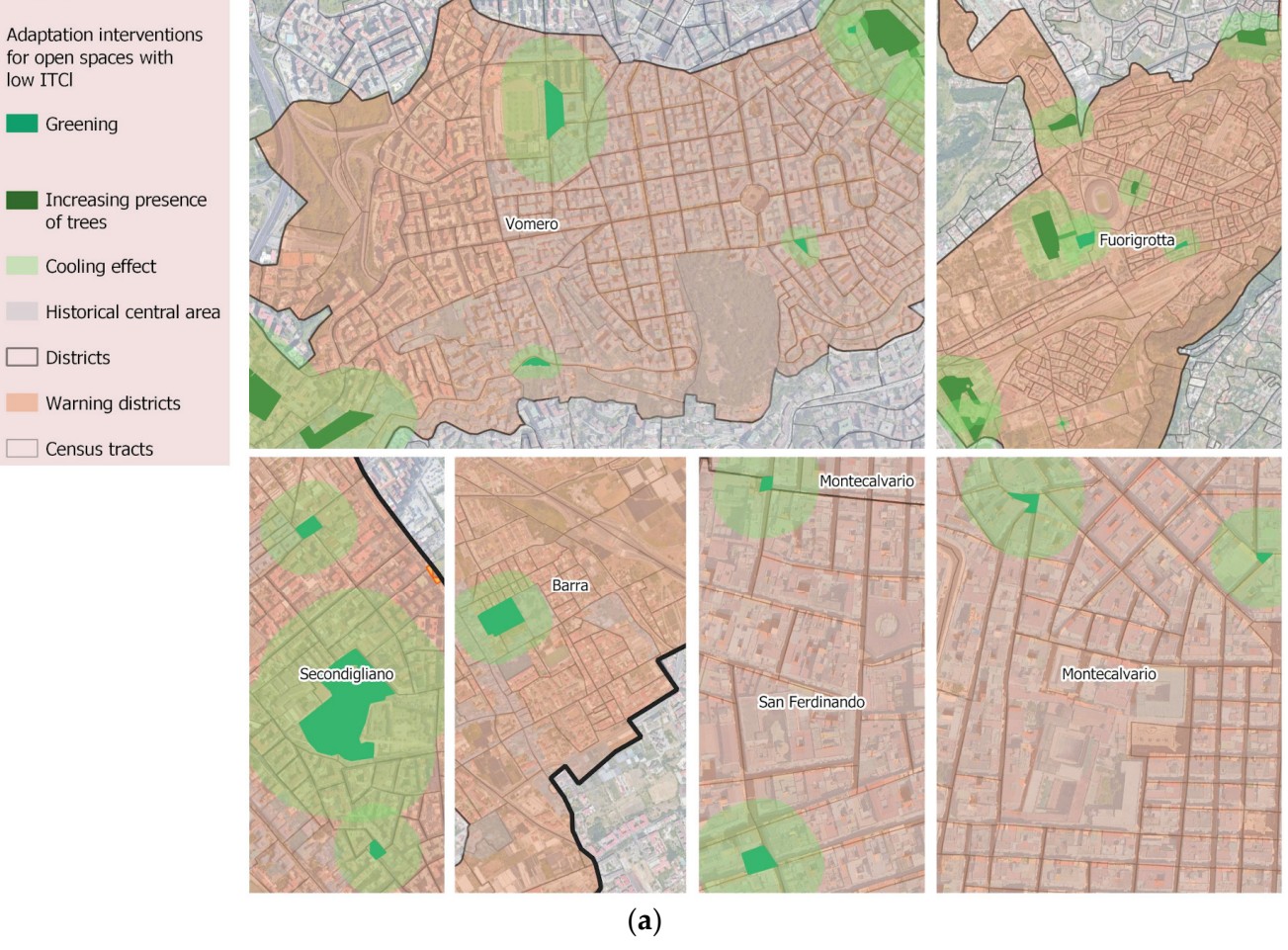

(a)

**Figure 7.** *Cont.*

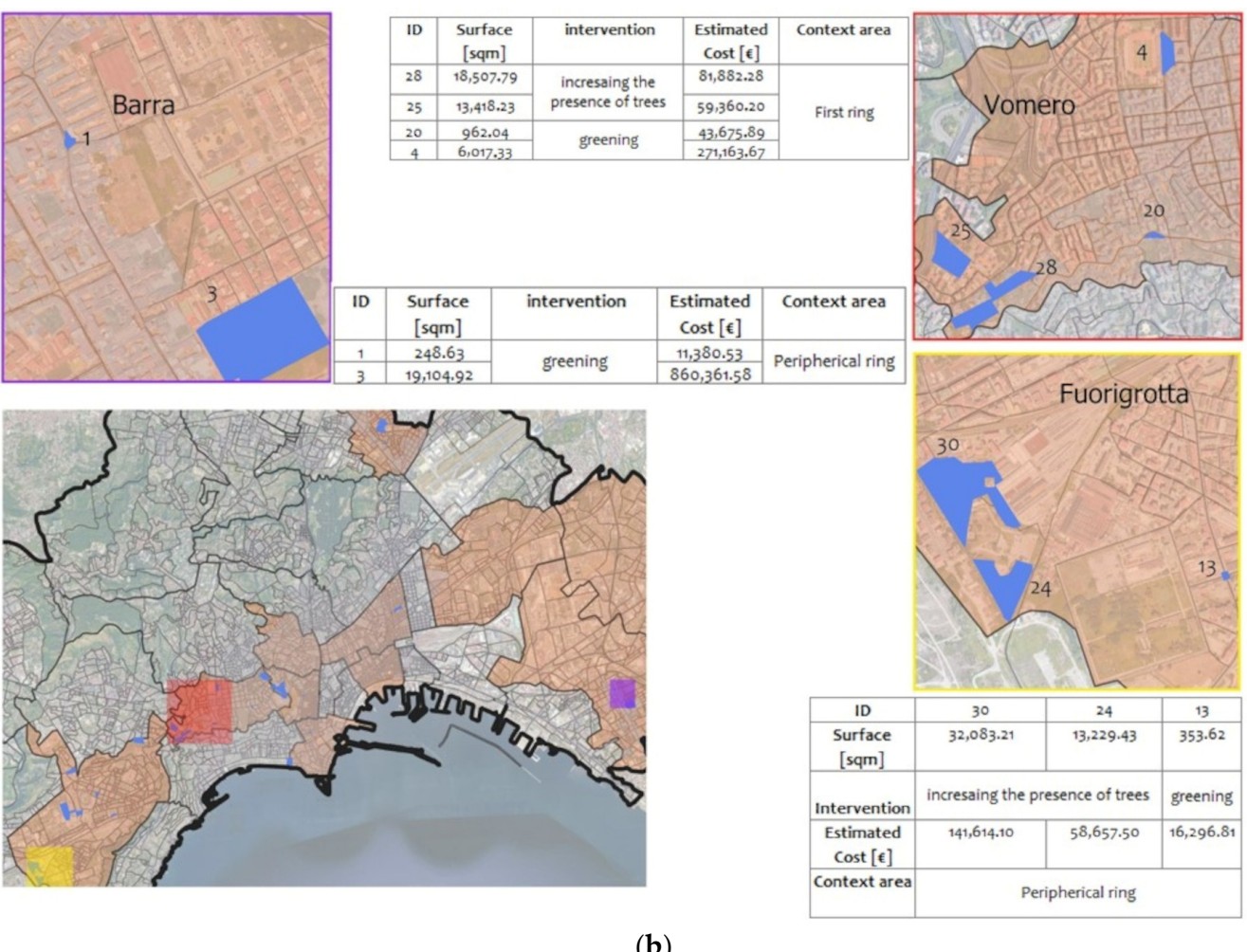

**(b)**

**Figure 7.** Classification of open spaces with local High Thermal Comfort Index in warning districts (**a**) and some examples of interventions and related costs of open spaces with a Low Permeability Index (**b**).

The proposed interventions require a total cost of just under 8 million euros and relate to both an increase in vegetation, to be implemented above all in the open spaces of Secondigliano, Barra and Ponticelli, and integrated solutions such as the creation of permeable surfaces and bioretention areas to support the existing drainage network by reducing runoff volumes and increasing the presence of vegetation above all in the open spaces of the peripheral districts such as Barra and Fuorigrotta (Figure S13).

In the open spaces located in the central urban area, the prevalent interventions are to increase vegetation and rain gardens, thanks to both the high $ITC_l$ values and the physical characteristics that guided the choice of interventions to be proposed. In particular, in the case of Piazza Municipio, it was decided to create a rain garden to further enhance the redevelopment of the open space now being completed and to act on improving permeability, given the presence of vegetation, albeit limited.

Remaining within the San Ferdinando district, it is worth noting that for the Molosiglio area, the work to strengthen the presence of trees fits in well with the redevelopment project for this green space located in the section of the promenade between the maritime station and the seafront, to contribute to increasing its attractiveness and usability, also by tourists, especially during the summer period of greatest thermal stress.

The cost estimate for adapting the open space system of the city of Naples to climate change seems to be higher for permeability improvement interventions due to the problem of the widespread vulnerability in different parts of the municipal territory and to the

consistency of the interventions requiring a greater degree of transformation of the space, compared to greening solutions which seem to be the least costly from an economic point of view.

### 5.4. Sorting Open Spaces According to Costs and Potential Benefits

Finally, the proposed adaptation measures were evaluated based on their effectiveness in carrying out a sorting useful to public decision-makers for the choice of open spaces to be transformed with priority. The sorting was carried out for each of the three climate vulnerability conditions considered (low $IP_l$ value, high $ICT_l$ value and co-presence of both "critical" values of both indices) and because of the two criteria underlying the study defined earlier (deviation criterion and cost-benefit criterion).

The orders were also defined by applying the Jenks algorithm that sets the limits between the various classes in correspondence with discontinuities or "jumps" in the distribution of values. In particular, this algorithm was applied as far as permeability is concerned, considering the size of the areas, and as far as thermal comfort is concerned, bearing in mind the cost per inhabitant. It was, thus, possible to define a first cluster of open spaces based on their size (defined by the largest jump in size in the Jenks sorting) and based on cost per inhabitant (defined by the significant jumps in the Jenks sorting), and a second cluster including all the other spaces (defined by the lack of significant jumps in the two sortings).

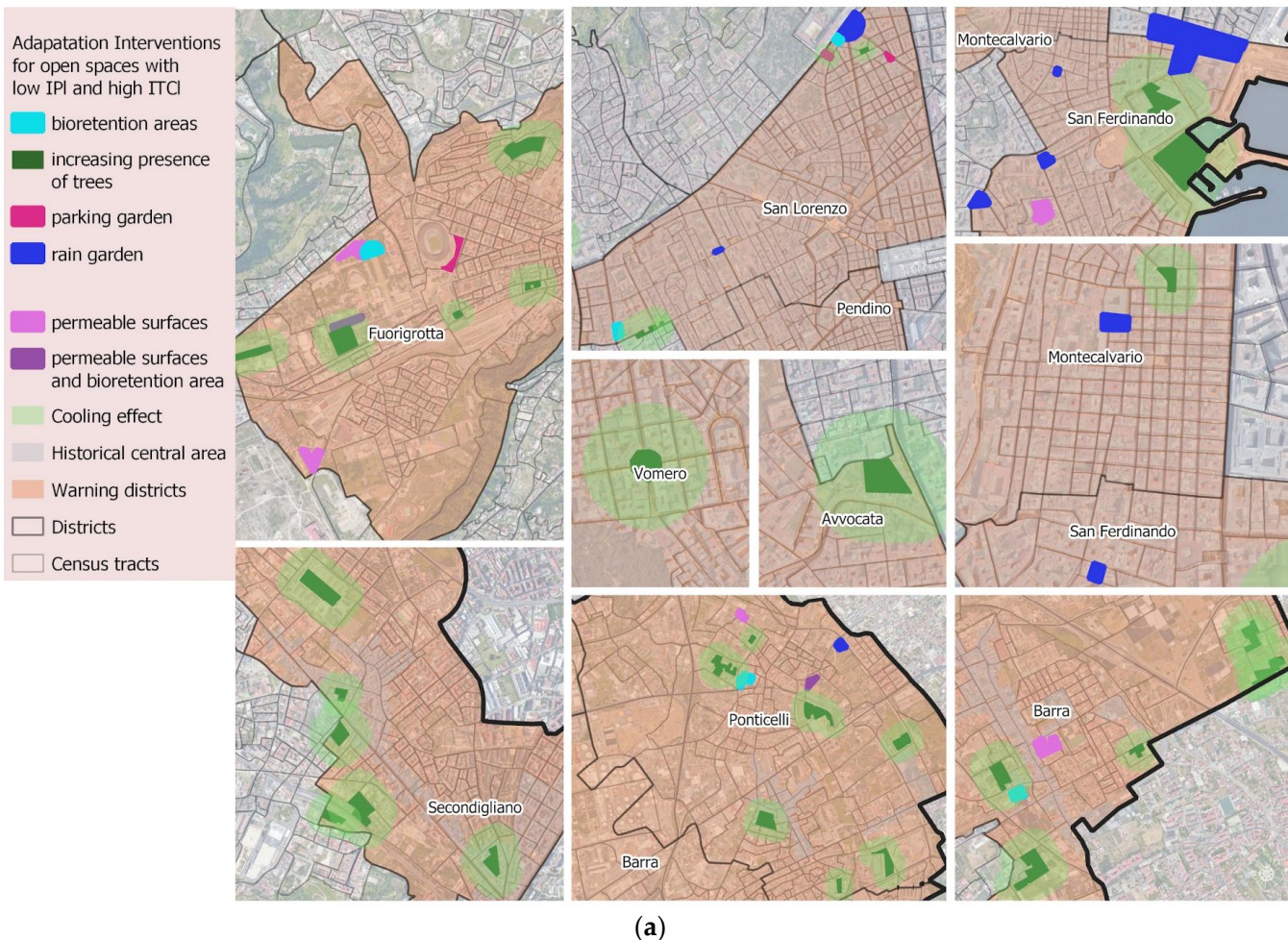

(**a**)

**Figure 8.** *Cont.*

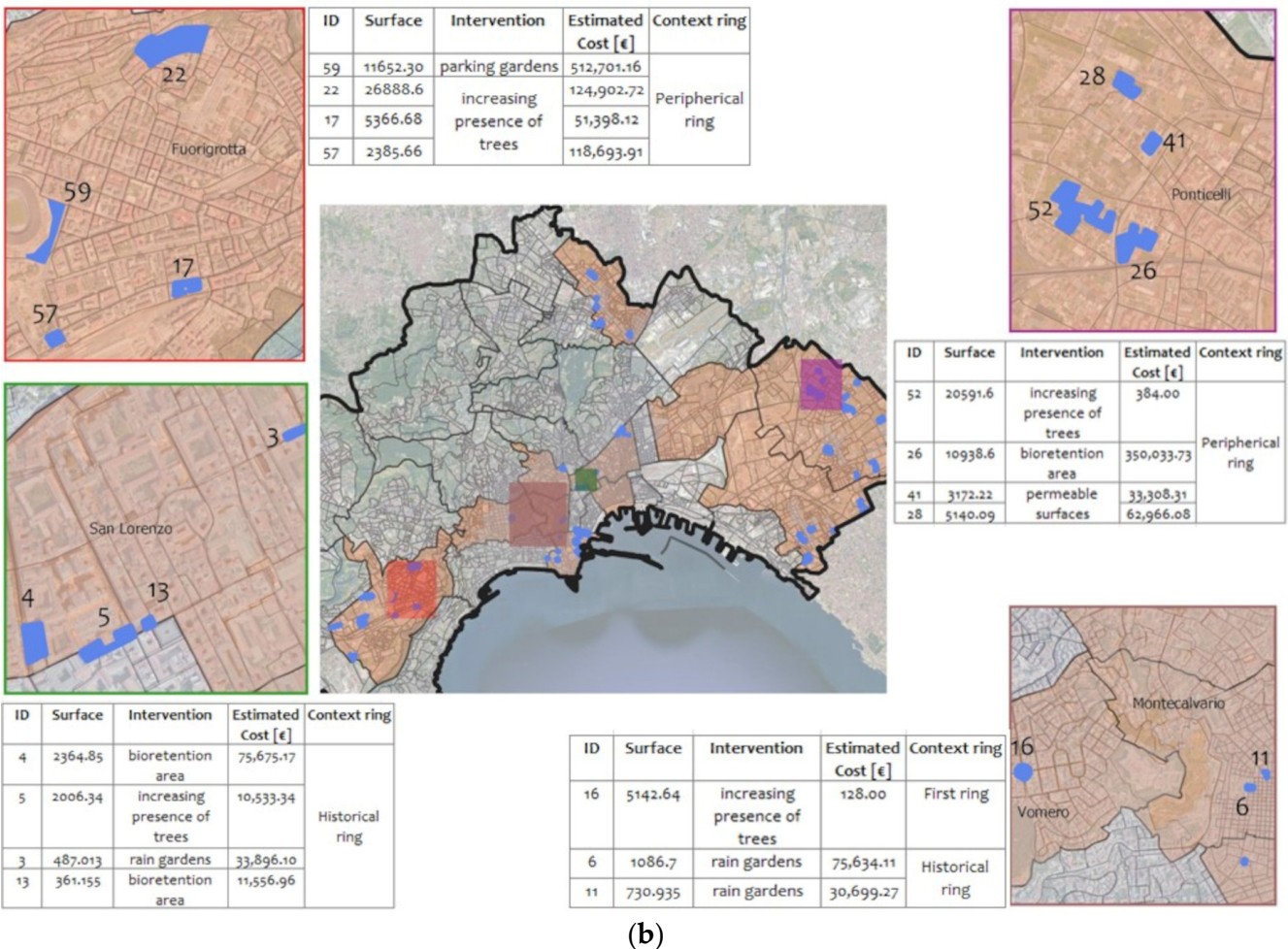

**(b)**

**Figure 8.** Classification of open spaces with local Low Permeability Index and local High Thermal Comfort Index in warning districts (**a**) and some examples of interventions and related costs of open spaces with a Low Permeability Index (**b**).

In detail, as far as permeability is concerned, the first cluster is composed of 11 open spaces located mainly in the peripheral area (Table 3), having both positive signs (identifying a low possibility of the economic burden of adaptation) and negative signs (identifying a high possibility of the economic burden of adaptation). Among the negative ones, two have a high delta signifying a significant economic burden related to the intervention of water squares. It is emphasised that all values with a positive sign refer to adaptation measures such as rain and parking gardens and bioretention areas to be realised in all areas of the city.

**Table 3.** Sorting and clusters of open spaces with low permeability, according to Δ values.

| ID | Surface [sqm] | District | Climate Adaption Intervention | Estimated Cost [€] | Context Area | Dimensional Weight of Open Space % | Economic Weight of Open Space % | Δ |
|---|---|---|---|---|---|---|---|---|
| 77 | 85,951.13 | Poggioreale | water square | 12,892,669.80 | Peripherical ring | 29.82% | 49.50% | −19.68% |
| 76 | 35,467.85 | Poggioreale | rain garden | 993,099.74 | Peripherical ring | 12.31% | 3.81% | 8.49% |
| 75 | 26,638.03 | Barra | water square | 3,995,704.50 | Peripherical ring | 9.24% | 15.34% | −6.10% |
| 74 | 24,565.03 | Fuorigrotta | parking garden | 1,621,292.18 | Peripherical ring | 8.52% | 6.22% | 2.30% |
| 73 | 8395.37 | Secondigliano | filter strips and rain garden | 990,653.42 | Peripherical ring | 2.91% | 3.80% | −0.89% |
| 72 | 8070.28 | San Lorenzo | bioretention areas | 258,249.02 | Historical ring | 2.80% | 0.99% | 1.81% |
| 71 | 7744.56 | Pendino | rain garden | 216,847.57 | Historical ring | 2.69% | 0.83% | 1.85% |

**Table 3.** *Cont.*

| ID | Surface [sqm] | District | Climate Adaption Intervention | Estimated Cost [€] | Context Area | Dimensional Weight of Open Space % | Economic Weight of Open Space % | Δ |
|---|---|---|---|---|---|---|---|---|
| 70 | 7394.53 | Vomero | bioretention areas and filter strips | 902,132.78 | First ring | 2.57% | 3.46% | −0.90% |
| 69 | 6321.09 | Poggioreale | bioretention areas | 202,274.88 | Peripherical ring | 2.19% | 0.78% | 1.42% |
| 68 | 4945.77 | Fuorigrotta | rain garden | 138,481.42 | Peripherical ring | 1.72% | 0.53% | 1.18% |
| 67 | 3770.04 | Poggioreale | bioretention areas | 120,641.34 | Peripherical ring | 1.31% | 0.46% | 0.84% |
| 66 | 3726.93 | Pendino | filter strips | 335,423.97 | Historical ring | 1.29% | 1.29% | 0.01% |
| 65 | 3463.71 | San Ferdinando | rain garden | 96,983.82 | Historical ring | 1.20% | 0.37% | 0.83% |
| 64 | 3228.01 | Fuorigrotta | rain garden | 90,384.22 | Peripherical ring | 1.12% | 0.35% | 0.77% |
| 63 | 2713.25 | Pendino | parking garden | 65,117.95 | Historical ring | 0.94% | 0.25% | 0.69% |
| 62 | 2179.46 | Avvocata | parking garden | 52,307.06 | Historical ring | 0.76% | 0.20% | 0.56% |
| 61 | 2069.04 | San Lorenzo | filter strips | 186,213.33 | Historical ring | 0.72% | 0.71% | 0.00% |
| 60 | 1944.51 | Fuorigrotta | bioretention areas | 62,224.22 | Peripherical ring | 0.67% | 0.24% | 0.44% |
| 59 | 1926.56 | Barra | rain garden | 53,943.74 | Peripherical ring | 0.67% | 0.21% | 0.46% |
| 58 | 1864.44 | Montecalvario | filter strips | 167,799.60 | Historical ring | 0.65% | 0.64% | 0.00% |
| 57 | 1839.17 | San Ferdinando | rain garden | 51,496.82 | Historical ring | 0.64% | 0.20% | 0.44% |
| 56 | 1837.22 | San Lorenzo | rain garden | 51,442.19 | Historical ring | 0.64% | 0.20% | 0.44% |
| 55 | 1775.09 | Avvocata | bioretention areas | 56,802.91 | Historical ring | 0.62% | 0.22% | 0.40% |
| 54 | 1707.33 | San Lorenzo | filter strips | 153,659.70 | Historical ring | 0.59% | 0.59% | 0.00% |
| 53 | 1643.94 | Avvocata | filter strips | 147,954.33 | Historical ring | 0.57% | 0.57% | 0.00% |
| 52 | 1639.47 | Pendino | rain garden | 45,905.08 | Historical ring | 0.57% | 0.18% | 0.39% |
| 51 | 1401.48 | Avvocata | filter strips | 126,133.02 | Historical ring | 0.49% | 0.48% | 0.00% |
| 50 | 1350.54 | Montecalvario | filter strips | 121,548.51 | Historical ring | 0.47% | 0.47% | 0.00% |
| 49 | 1304.90 | Pendino | parking garden | 54,805.93 | Historical ring | 0.45% | 0.21% | 0.24% |
| 48 | 1273.84 | Avvocata | rain garden | 35,667.60 | Historical ring | 0.44% | 0.14% | 0.31% |
| 47 | 1213.14 | San Lorenzo | rain garden | 33,967.78 | Historical ring | 0.42% | 0.13% | 0.29% |
| 46 | 1173.49 | San Lorenzo | parking garden | 49,286.50 | Historical ring | 0.41% | 0.19% | 0.22% |
| 45 | 1155.04 | Pendino | filter strips | 103,953.69 | Historical ring | 0.40% | 0.40% | 0.00% |
| 44 | 1100.03 | San Lorenzo | rain garden | 30,800.73 | Historical ring | 0.38% | 0.12% | 0.26% |
| 43 | 1099.31 | Vomero | filter strips | 98,937.81 | First ring | 0.38% | 0.38% | 0.00% |
| 42 | 1012.56 | Montecalvario | rain garden | 28,351.62 | Historical ring | 0.35% | 0.11% | 0.24% |
| 41 | 924.17 | San Lorenzo | filter strips | 83,175.57 | Historical Area | 0.32% | 0.32% | 0.00% |
| 40 | 906.62 | Avvocata | filter strips | 81,595.62 | Historical ring | 0.31% | 0.31% | 0.00% |
| 39 | 891.25 | San Lorenzo | rain garden | 24,955.00 | Historical ring | 0.31% | 0.10% | 0.21% |
| 38 | 887.06 | San Lorenzo | filter strips | 79,835.58 | Historical ring | 0.31% | 0.31% | 0.00% |
| 37 | 885.77 | Pendino | rain garden | 24,801.62 | Historical ring | 0.31% | 0.10% | 0.21% |
| 36 | 874.73 | Vomero | bioretention areas | 41,987.14 | First ring | 0.30% | 0.16% | 0.14% |
| 35 | 823.89 | Pendino | filter strips | 74,150.19 | Historical ring | 0.29% | 0.28% | 0.00% |
| 34 | 799.41 | Montecalvario | rain garden | 22,383.59 | Historical ring | 0.28% | 0.09% | 0.19% |
| 33 | 780.34 | San Lorenzo | filter strips | 70,230.96 | Historical ring | 0.27% | 0.27% | 0.00% |
| 32 | 770.35 | Montecalvario | rain garden | 21,569.86 | Historical ring | 0.27% | 0.08% | 0.18% |
| 31 | 768.70 | Vomero | bioretention areas | 36,897.50 | First ring | 0.27% | 0.14% | 0.13% |
| 30 | 732.06 | Pendino | filter strips | 65,885.49 | Historical ring | 0.25% | 0.25% | 0.00% |
| 29 | 726.68 | Vomero | rain garden | 20,347.04 | First ring | 0.25% | 0.08% | 0.17% |
| 28 | 662.57 | Pendino | filter strips | 59,631.03 | Historical ring | 0.23% | 0.23% | 0.00% |
| 27 | 657.87 | Pendino | parking garden | 27,630.37 | Historical ring | 0.23% | 0.11% | 0.12% |

Table 3. *Cont.*

| ID | Surface [sqm] | District | Climate Adaption Intervention | Estimated Cost [€] | Context Area | Dimensional Weight of Open Space % | Economic Weight of Open Space % | Δ |
|----|------|----------|---------|---------|--------------|-------|-------|-------|
| 26 | 646.03 | Barra | rain garden | 18,088.92 | Peripherical ring | 0.22% | 0.07% | 0.15% |
| 25 | 638.52 | San Lorenzo | filter strips | 57,466.98 | Historical ring | 0.22% | 0.22% | 0.00% |
| 24 | 638.44 | Pendino | filter strips | 57,459.78 | Historical ring | 0.22% | 0.22% | 0.00% |
| 23 | 615.56 | Pendino | rain garden | 17,235.79 | Historical ring | 0.21% | 0.07% | 0.15% |
| 22 | 577.55 | Avvocata | bioretention areas | 27,722.35 | Historical ring | 0.20% | 0.11% | 0.09% |
| 21 | 565.29 | Pendino | filter strips | 50,876.37 | Historical ring | 0.20% | 0.20% | 0.00% |
| 20 | 558.67 | Pendino | rain garden | 15,642.73 | Historical ring | 0.19% | 0.06% | 0.13% |
| 19 | 494.87 | San Ferdinando | filter strips | 44,537.94 | Historical ring | 0.17% | 0.17% | 0.00% |
| 18 | 474.03 | San Lorenzo | filter strips | 42,662.52 | Historical ring | 0.16% | 0.16% | 0.00% |
| 17 | 468.77 | Pendino | filter strips | 42,189.39 | Historical ring | 0.16% | 0.16% | 0.00% |
| 16 | 468.55 | San Lorenzo | filter strips | 42,169.14 | Historical ring | 0.16% | 0.16% | 0.00% |
| 15 | 465.30 | Pendino | rain garden | 13,028.40 | Historical ring | 0.16% | 0.05% | 0.11% |
| 14 | 453.73 | Pendino | rain garden | 12,704.47 | Historical ring | 0.16% | 0.05% | 0.11% |
| 13 | 445.28 | Avvocata | bioretention areas | 21,373.44 | Historical ring | 0.15% | 0.08% | 0.07% |
| 12 | 404.69 | San Lorenzo | parking garden | 16,997.15 | Historical ring | 0.14% | 0.07% | 0.08% |
| 11 | 396.45 | Pendino | rain garden | 11,100.66 | Historical ring | 0.14% | 0.04% | 0.09% |
| 10 | 392.90 | Montecalvario | rain garden | 11,001.28 | Historical ring | 0.14% | 0.04% | 0.09% |
| 9 | 386.61 | Montecalvario | filter strips | 34,794.99 | Historical ring | 0.13% | 0.13% | 0.00% |
| 8 | 366.33 | Vomero | bioretention areas | 17,583.89 | First ring | 0.13% | 0.07% | 0.06% |
| 7 | 329.61 | Avvocata | filter strips | 29,664.90 | Historical ring | 0.11% | 0.11% | 0.00% |
| 6 | 303.21 | Pendino | rain garden | 8489.91 | Historical ring | 0.11% | 0.03% | 0.07% |
| 5 | 302.75 | Avvocata | rain garden | 8477.08 | Historical ring | 0.11% | 0.03% | 0.07% |
| 4 | 284.39 | San Lorenzo | filter strips | 25,594.83 | Historical ring | 0.10% | 0.10% | 0.00% |
| 3 | 263.79 | Montecalvario | filter strips | 23,741.37 | Historical ring | 0.09% | 0.09% | 0.00% |
| 2 | 188.73 | Montecalvario | filter strips | 16,985.25 | Historical ring | 0.07% | 0.07% | 0.00% |
| 1 | 109.65 | San Lorenzo | filter strips | 9868.41 | Historical ring | 0.04% | 0.04% | 0.00% |

The second cluster consists of open spaces located mainly in the central districts of Avvocata, San Lorenzo and Pendino, characterised by more limited burdens (Table 3).

Within this cluster, it is possible to define groups of open spaces classified according to the type of intervention and the urban sector (peripheral crown, first crown, central crown) in which it falls to provide the local decision maker with further useful elements for choosing how and where to intervene (Figures S14 and S15). For example, rain gardens and filter strips are the most widespread interventions to improve permeability, which especially lack in the peripheral area (Figures S14 and S15).

As far as thermal comfort is concerned, the effectiveness was evaluated concerning the cost per inhabitant of the greening intervention to be carried out due to the results of the previous work developed by the authors. Again, two clusters were identified. The first is made up of four open spaces (characterised by a higher economic burden to bear) located in the peripheral districts of Fuorigrotta, Secondigliano and Barra (Table 4). The second cluster consists of 27 open spaces located in more densely populated areas, which entail a lower unit cost of just over 150 euros.

Depending on the type of intervention and the urban sector (peripheral crown, first crown, central crown) in which each open space falls, greening interventions are needed in the most stratified areas, while in the peripheral areas, interventions aimed at improving thermal comfort are needed (Figures S16 and S17).

**Table 4.** Sorting and clusters of open spaces with high thermal comfort, according to cost per inhabitant values.

| ID | Surface [sqm] | District | Climate Adaption Intervention | Estimated Cost [€] | Context Area | Cost Per Inhab. | Inhab. |
|----|--------------|----------|------------------------------|--------------------|--------------|-----------------|--------|
| 19 | 31,273.59 | Secondigliano | greening | 1408.271.42 | Peripherical ring | 1635.62 | 861 |
| 7 | 8962.93 | Fuorigrotta | greening | 403,907.63 | Peripherical ring | 585.37 | 690 |
| 31 | 39,463.07 | Fuorigrotta | increasing the presence of trees | 174,085.51 | Peripherical ring | 391.20 | 445 |
| 3 | 19,104.92 | Barra | greening | 860,361.58 | Peripherical ring | 375.38 | 2292 |
| 6 | 13,416.13 | San Ferdinando | greening | 603,981.85 | Historical ring | 150.88 | 4003 |
| 29 | 30,742.18 | Vomero | increasing the presence of trees | 135,713.57 | First ring | 110.61 | 1227 |
| 4 | 6017.33 | Vomero | greening | 271,163.67 | First ring | 82.05 | 3305 |
| 28 | 18,507.79 | Vomero | increasing the presence of trees | 81,882.28 | First ring | 56.78 | 1442 |
| 8 | 2534.14 | Vomero | greening | 114,420.35 | First ring | 53.59 | 2135 |
| 30 | 32,083.21 | Fuorigrotta | increasing the presence of trees | 141,614.10 | Peripherical ring | 45.22 | 3132 |
| 20 | 962.04 | Vomero | greening | 43,675.89 | First ring | 36.52 | 1196 |
| 9 | 1932.27 | Fuorigrotta | greening | 87,335.93 | Peripherical ring | 27.21 | 3210 |
| 26 | 13,442.93 | Vomero | increasing the presence of trees | 59,596.89 | First ring | 26.87 | 2218 |
| 5 | 653.90 | Vomero | greening | 29,809.59 | First ring | 26.08 | 1143 |
| 27 | 17,594.65 | Fuorigrotta | increasing the presence of trees | 77,864.46 | Peripherical ring | 24.79 | 3141 |
| 17 | 1733.07 | Secondigliano | greening | 78,372.33 | Peripherical ring | 23.81 | 3292 |
| 23 | 10,771.52 | Fuorigrotta | increasing the presence of trees | 47,842.70 | Peripherical ring | 22.36 | 2140 |
| 18 | 1.606,77 | Secondigliano | greening | 72,688.52 | Peripherical ring | 21.70 | 3350 |
| 25 | 13,418.23 | Vomero | increasing the presence of trees | 59,360.20 | First ring | 18.84 | 3150 |
| 16 | 1150.11 | Secondigliano | greening | 52,139.13 | Peripherical ring | 15.90 | 3280 |
| 24 | 13,229.43 | Fuorigrotta | increasing the presence of trees | 58,657.50 | Peripherical ring | 14.62 | 4011 |
| 15 | 339.53 | San Ferdinando | greening | 15,342.90 | Historical ring | 7.15 | 2145 |
| 13 | 353.62 | Fuorigrotta | greening | 16,296.81 | Peripherical ring | 6.67 | 2442 |
| 21 | 2841.12 | Fuorigrotta | increasing the presence of trees | 12,820.92 | Peripherical ring | 5.78 | 2220 |
| 22 | 3779.97 | San Lorenzo | increasing the presence of trees | 16,951.86 | Historical ring | 5.17 | 3281 |
| 2 | 289.28 | Vomero | greening | 13,209.78 | First ring | 4.14 | 3188 |
| 12 | 78.02 | San Ferdinando | greening | 3894.81 | Historical ring | 3.55 | 1098 |
| 1 | 248.63 | Barra | greening | 11,380.53 | Peripherical ring | 3.20 | 3555 |
| 11 | 59.11 | Montecalvario | greening | 3043.73 | Historical ring | 1.40 | 2172 |
| 14 | 94.05 | Montecalvario | greening | 4296.25 | Historical ring | 1.38 | 3122 |
| 10 | 52.53 | Montecalvario | greening | 2428.03 | Historical ring | 0.77 | 3155 |

A third result of the work is the identification of spaces that need contextual adaptation to the two types of vulnerability (Table 5).

**Table 5.** Sorting and clusters of open spaces with low permeability and high thermal comfort, according to Δ values and cost per inhabitant values.

| ID | Surface [sqm] | District | Climate Adaption Intervention | Estimated Cost [€] | Context Ring | Dimensional Weight of Open Space % | Economic Weight of Open Space % | Δ | Cost Per Inhab. | Inhab. |
|---|---|---|---|---|---|---|---|---|---|---|
| 1 | 45,010.06 | San Ferdinando | rain gardens | 1,890,422.52 | Historical ring | 13.3% | 43.0% | −29.7% | | |
| 29 | 14,983.65 | San Ferdinando | permeable surfaces | 183,549.71 | Historical ring | 4.4% | 4.2% | 0.3% | | |
| 21 | 13,299.43 | Fuorigrotta | permeable surfaces | 162,918.03 | Peripherical ring | 3.9% | 3.7% | 0.2% | | |
| 20 | 13,229.43 | Fuorigrotta | bioretention area | 423,341.79 | Peripherical ring | 3.9% | 9.6% | −5.7% | | |
| 30 | 12,329.24 | Fuorigrotta | bioretention area | 394,535.74 | Peripherical ring | 3.7% | 9.0% | −5.3% | | |
| 59 | 11,652.30 | Fuorigrotta | parking gardens | 512,701.16 | Peripherical ring | 3.5% | 11.7% | −8.2% | | |
| 27 | 11,538.52 | Fuorigrotta | permeable surfaces | 141,346.92 | Peripherical ring | 3.4% | 3.2% | 0.2% | | |
| 47 | 11,518.50 | Barra | permeable surfaces | 120,944.22 | Peripherical ring | 3.4% | 2.8% | 0.7% | | |
| 26 | 10,938.55 | Ponticelli | bioretention area | 350,033.73 | Peripherical ring | 3.2% | 8.0% | −4.7% | | |
| 46 | 9831.81 | San Ferdinando | increasing presence of trees | 51,065.39 | Historical ring | | | | 1021.31 | 50 |
| 35 | 25,801.66 | Ponticelli | increasing presence of trees | 48,513.64 | Peripherical ring | | | | 312.99 | 155 |
| 39 | 13,526.49 | Fuorigrotta | increasing presence of trees | 43,643.95 | Peripherical ring | | | | 256.73 | 170 |
| 19 | 8822.32 | Fuorigrotta | permeable surfaces and bioretention area | 727,841.32 | Peripherical ring | 2.6% | 16.6% | −13.9% | | |
| 45 | 7289.34 | Ponticelli | rain gardens | 306,152.36 | Peripherical ring | 2.2% | 7.0% | −4.8% | | |
| 36 | 5753.30 | Ponticelli | permeable surfaces and bioretention area | 312,116.47 | Peripherical ring | 1.7% | 7.1% | −5.4% | | |
| 28 | 5140.09 | Ponticelli | permeable surfaces | 62,966.08 | Peripherical ring | 1.5% | 1.4% | 0.1% | | |
| 43 | 4840.49 | Barra | permeable surfaces and bioretention area | 166,996.94 | Peripherical ring | 1.4% | 3.8% | −2.4% | | |
| 41 | 3172.22 | Barra | permeable surfaces | 33,308.31 | Peripherical ring | 0.9% | 0.8% | 0.2% | | |
| 4 | 2364.85 | San Lorenzo | bioretention area | 75,675.17 | Historical ring | 0.7% | 1.7% | −1.0% | | |
| 9 | 2021.07 | San Ferdinando | rain gardens | 140,666.33 | Historical ring | 0.6% | 3.2% | −2.6% | | |
| 7 | 1182.67 | San Lorenzo | bioretention area | 37,845.47 | Historical ring | 0.4% | 0.9% | −0.5% | | |
| 6 | 1086.70 | Montecalvario | rain gardens | 75,634.11 | Historical ring | 0.3% | 1.7% | −1.4% | | |
| 15 | 844.99 | San Lorenzo | bioretention area | 27,039.74 | Historical ring | 0.3% | 0.6% | −0.4% | | |
| 11 | 730.94 | Montecalvario | rain gardens | 30,699.27 | Historical ring | 0.2% | 0.7% | −0.5% | | |
| 10 | 545.40 | San Ferdinando | rain gardens | 37,960.05 | Historical ring | 0.2% | 0.9% | −0.7% | | |
| 2 | 537.95 | San Lorenzo | parking gardens | 23,669.76 | Historical ring | 0.2% | 0.5% | −0.4% | | |
| 3 | 487.01 | San Lorenzo | rain gardens | 33,896.10 | Historical ring | 0.1% | 0.8% | −0.6% | | |
| 13 | 361.16 | San Lorenzo | bioretention area | 11,556.96 | Historical ring | 0.1% | 0.3% | −0.2% | | |
| 44 | 28,357.89 | San Ferdinando | v presence of trees | 837.46 | Historical ring | | | | 20.94 | 40 |
| 37 | 28,089.60 | Fuorigrotta | increasing presence of trees | 837.46 | Peripherical ring | | | | 111.46 | 814 |
| 22 | 26,888.62 | Fuorigrotta | increasing presence of trees | 124,902.72 | Peripherical ring | | | | 105.40 | 1185 |
| 40 | 23,053.21 | Ponticelli | increasing presence of trees | 118,693.91 | Peripherical ring | | | | 73.03 | 195 |
| 55 | 21,023.59 | Barra | increasing presence of trees | 113,911.31 | Peripherical ring | | | | 61.45 | 1118 |
| 52 | 20,591.57 | Ponticelli | increasing presence of trees | 101,818.10 | Peripherical ring | | | | 60.31 | 1968 |
| 50 | 16,777.54 | Ponticelli | increasing presence of trees | 92,887.80 | Peripherical ring | | | | 57.56 | 1979 |
| 24 | 15,527.81 | Ponticelli | increasing presence of trees | 90,730.89 | Peripherical ring | | | | 41.45 | 2241 |
| 33 | 13,466.58 | Secondigliano | increasing presence of trees | 68,706.36 | Peripherical ring | | | | 41.12 | 954 |
| 34 | 13,466.58 | Secondigliano | increase presence of trees | 59,900.54 | Peripherical ring | | | | 39.56 | 2574 |
| 48 | 11,518.53 | Barra | increasing presence of trees | 59,636.94 | Peripherical ring | | | | 37.89 | 1574 |
| 8 | 10,536.70 | San Lorenzo | increasing presence of trees | 59,380.94 | Historical ring | | | | 35.16 | 385 |
| 18 | 8827.36 | Barra | increasing presence of trees | 46,489.49 | Peripherical ring | | | | 22.75 | 1009 |
| 38 | 7640.75 | Secondigliano | increasing presence of trees | 15,473.50 | Peripherical ring | | | | 21.16 | 986 |
| 44 | 28,357.89 | San Ferdinando | increase presence of trees | 837.46 | Historical ring | | | | 20.94 | 40 |
| 31 | 7058.82 | Avvocata | increasing presence of trees | 39,224.38 | Historical ring | | | | 20.52 | 968 |
| 51 | 6792.21 | Secondigliano | increase presence of trees | 34,003.29 | Peripherical ring | | | | 19.85 | 3017 |
| 49 | 5766.92 | Barra | increase presence of trees | 31,442.79 | Peripherical ring | | | | 19.53 | 3041 |

**Table 5.** *Cont.*

| ID | Surface [sqm] | District | Climate Adaption Intervention | Estimated Cost [€] | Context Ring | Dimensional Weight of Open Space % | Economic Weight of Open Space % | Δ | Cost Per Inhab. | Inhab. |
|---|---|---|---|---|---|---|---|---|---|---|
| 17 | 5366.68 | Fuorigrotta | increasing presence of trees | 30,013.74 | Peripherical ring | | | | 18.30 | 1245 |
| 16 | 5142.64 | Vomero | increasing presence of trees | 25,502.45 | First ring | | | | 17.03 | 1846 |
| 53 | 5129.80 | Ponticelli | increase presence of trees | 23,997.37 | Peripherical ring | | | | 15.00 | 3100 |
| 54 | 5129.80 | Ponticelli | increasing presence of trees | 23,011.59 | Peripherical ring | | | | 13.78 | 2178 |
| 23 | 4683.99 | Secondigliano | increase presence of trees | 22,955.12 | Peripherical ring | | | | 11.33 | 3001 |
| 32 | 4427.60 | Barra | increasing presence of trees | 22,782.72 | Peripherical ring | | | | 11.29 | 1270 |
| 25 | 3755.27 | Barra | increase presence of trees | 20,865.56 | Peripherical ring | | | | 9.96 | 2310 |
| 42 | 3172.22 | Ponticelli | increasing presence of trees | 19,865.42 | Peripherical ring | | | | 8.95 | 2682 |
| 14 | 3149.29 | San Ferdinando | increasing presence of trees | 16,651.19 | Historical ring | | | | 7.72 | 2005 |
| 58 | 3047.32 | Secondigliano | increasing presence of trees | 14,341.77 | Peripherical ring | | | | 6.37 | 4001 |
| 57 | 2385.66 | Fuorigrotta | increasing presence of trees | 14,240.88 | Peripherical ring | | | | 5.08 | 165 |
| 5 | 2006.34 | San Lorenzo | increasing presence of trees | 13,536.22 | Historical ring | | | | 4.53 | 3677 |
| 12 | 1395.41 | San Lorenzo | increasing presence of trees | 10,624.90 | Historical ring | | | | 3.67 | 2898 |

Therefore, two clusters were identified based on both the delta between dimensional weight and economic weight (for permeability) and the cost per inhabitant (for thermal comfort). The first cluster is represented by open spaces with a negative delta, i.e., with the highest economic burden to be borne, relative to rain garden interventions (in the San Ferdinando district) and parking garden and bioretention area interventions (in the Fuorigrotta district); by open spaces with numerically lower positive deltas (e.g., Barra neighbourhood, Table 5); by open spaces with the highest cost per inhabitant relative to increases in trees (in the San Ferdinando neighbourhood) and very large open spaces (14,000 sqm) in a non-densely inhabited area (in the Ponticelli and Fuorigrotta neighbourhoods).

Interventions to increase vegetation are the most numerous for the reduction of the dual climatic vulnerability that characterises the open spaces under consideration (Figure S18). The peripheral part of the city is the one where the open spaces with low thermal and hydraulic performance are mainly located (Figure S19); however, it is worth noting that the central crown is also strongly characterised by this dual problem, confirming the need to update the transformation rules of areas of high historical-architectural value with the criteria that take into account the resilience essentials cities require.

## 6. Conclusions

Climate change is a long-term challenge, but the pace and intensity of its effects that affect cities all over the planet require urgent and innovative strategies not only in the "mitigation" of the phenomena but, above all, in the "adaptation" of cities to the growing impacts of these new climatic events. This is even more true for the urban systems of the countries bordering the Mediterranean, which are threatened by the effects of global warming. In this geographic area, a large part of the population lives in coastal areas, which are more exposed and vulnerable to the natural phenomena associated with climate change [63,64].

To rapidly adapt cities to new climatic conditions, reducing their vulnerability to likely negative impacts, the open space system can play a relevant role in cooling the built environment, improving stormwater drainage and promoting sustainable mobility. These spaces can be assigned important climate-regulating functions as drivers and accelerators of sustainable urban development, urban regeneration and resilient systems [65–67].

From this perspective, this contribution represents the first result of a work aimed at developing a decision-support tool to sustainably transform the open space system (built and unbuilt spaces) by reducing its vulnerability and increasing its attractiveness and urban quality.

The application of the proposed method to the urban scale allows for (i) obtaining an initial cognitive result of the system of open spaces in terms of their characteristics (physical, climatic and usability) and the external agents by which they may be affected (such as heat waves and extreme rainfall events) and (ii) outlining some possible adaptation strategies in the different parts of the city that also take into account the resources required for their implementation. The estimated costs, the type of intervention proposed and the urban reference context represent three possible elements for local decision-makers to validate/choose the climate change adaptation interventions to be implemented. The results can represent useful inputs to support the development of climate adaptation strategy and plan at the urban scale that is strongly needed in populous and built densely cities like Naples, where chronic social and economic issues can be exacerbated by the increase of frequency and intensity of extreme precipitation events and heat waves, representing a signal of the ongoing climate change [10].

The low performance of the open spaces to extreme climatic events, like flooding and UHI, is mostly due to the high imperviousness and building density levels of the city. Two hundred and seventy-nine open spaces out of a total of four hundred and forty-five require adaptation interventions with a higher financial burden for permeability improvement, which underlines the relevance of the issue of land use in relation to sustainability and urban resilience. The context of the densely built and stratified city, where the need to adapt the physical, functional and architectural heritage is bound to clash, inevitably, with the immobility of transformations determined by urban planning and building rules and regulations, makes the results significant for the Italian panorama.

The proposed work, in the next step of the research, will allow defining intervention practices that, according to the preservation of a city's historical heritage, will stimulate the sensitivity of local administrators in innovating the urban planning tools in force in the light of current climatic-energy requirements.

To the best of our knowledge, the present study is the first to provide climate adaptation interventions based on NBSs and relative estimated costs for the Naples case study and, representing an initial result, in a subsequent phase of work, it will be necessary to measure the weight of the relationships between context characteristics and open spaces and to define the "connection network" between open spaces and the set of adaptation interventions for each space. Further applications will rely on flood and microclimate simulations to assess the hydrological and thermal suitability effects of the proposed interventions. Different scenarios can be simulated to measure and compare consequent benefits in terms of reduction of temperature and runoff coefficient and level of pluvial floods. Through these future developments, the following current limits of the work can be overcome: (i) the lack of data on the flooded surfaces, as they require hydrologic models and sewer system information; (ii) the statistical significance of the variables to assess the influence of the territorial context on permeability and thermal comfort properties of the open spaces; (iii) the data related to microclimate changes consequent to the realisation of the proposed NBSs.

**Supplementary Materials:** The following supporting information can be downloaded at: https://www.mdpi.com/article/10.3390/su15108111/s1, The supplementary materials contains additional maps and tables related to the classification of open spaces and neighbourhoods based on (i) physical characteristics that are relevant to climate vulnerability and (ii) estimated costs and benefits determined by proposed climate adaptation interventions. Figure S1. Classification of open spaces based on Runoff coefficients. Figure S2. Classification of open spaces based on cooling effect, according to green areas dimensions and urban fabrics building density. Figure S3. Classification of open spaces based on Pedestrian accessibility. Figure S4. Classification of open spaces based on LPT accessibility. Figure S5. Classification of open spaces based on Parking supply accessibility. Figure S6. Land Surface Temperature ($30 \times 30$ m grid). Figure S7. Classification of districts based on UHI levels. Figure S8. Amount of natural surfaces within districts and their classification based on $IP_t$. Figure S9. Classification of districts based on Building density. Figure S10. Naples neighbourhoods representing "warning areas" due to their permeability and thermal comfort values indexes.

Figure S11. Proposed adaptation interventions for increasing permeability and related estimated costs. Figure S12. Proposed adaptation interventions for improving thermal comfort and related estimated costs. Figure S13. Proposed adaptation interventions for improving permeability and thermal comfort and related estimated costs. Figure S14. Classification of open spaces, according to adaptation interventions for improving permeability. Figure S15. Classification of open spaces for improving permeability, according to their district localisation. Figure S16. Classification of open spaces, according to adaptation interventions for improving thermal comfort. Figure S17. Classification of open spaces for improving thermal comfort, according to their district localisation. Figure S18. Classification of open spaces according to adaptation interventions for improving permeability and thermal comfort. Figure S19. Classification of open spaces for improving permeability and thermal comfort, according to their district localisation.

**Author Contributions:** Conceptualization, C.G.; Methodology, **C.G. and F.Z.**; Formal analysis, F.Z.; Investigation, F.Z.; Writing—original draft, F.Z.; Writing—review & editing, C.G.; Supervision, C.G. All authors have read and agreed to the published version of the manuscript.

**Funding:** This research received no external funding.

**Informed Consent Statement:** Not applicable.

**Data Availability Statement:** Data sharing not applicable.

**Conflicts of Interest:** The authors declare no conflict of interest.

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
