# Peer review of "A Method Proposal to Adapt Urban Open-Built and Green Spaces to Climate Change"

_sustainability, doi:10.3390/su15108111_

Round 1
Reviewer 1 Report
1. Improve the structure of the paper as it should include sections on
- Introduction
- Materials and Methods
- Results
- Discussion
- Conclusions
as it is advised in the Sustainability Instructions for authors.
2. Revise the Introduction section by making the necessary link between the green spaces concept, which is highlighted in this specific section and the open spaces concept, which is included in the other chapters. Change the research question accordingly and include the objectives of the paper. Then make the necessary linkage with the Conclusion whoch states that is the first to provide NBSs.
3. Explain why the focus of paper is only on the 2 types of extreme events: flooding and UHI
4. Please describe the acronyms on LST, LPT. Check the formulas 1-5 and explain how the components can be assembled, for example the runoff coefficient with permeable surface and UHI; UHI with cooling area; UHI with building density; runoff coefficient with natural surface and slope etc.
5. The steps of methodology may be better emphasized in a graphical representation.
6. include a reference for the 352-354 (for the statement that below 55sqm is no longer a space but an element of street furniture
7. Explain what is the total permeability index and total Thermal Comfort Index – Figures 5-6
8. Please make the paper more reader-friendly, the numerous figures that take up a page each are not easy to follow.
9. Explain if the interventions in Figure 11 are already perfomed and if it is so, explain their relevance for the paper and describe them properly. Explain the figure of 8 million euro in line 577.
10. Figure 14 is a table and is totally unattractive, please find another solution to communicate the information. Same for Figure 15 where you also should correct the word ’incresaing’ with ’increasing’. Same for Figure 16 where you should also complete the information for ’increasing the presence of’. Of what?
11. Please include arguments and discussion of findings which have to be coherent, balanced and compelling
12. Conclusion should also include the Limitations of the paper.
13. Please correlate the Conclusion with the paper content, especially in the line 703-704 which state that relative estimated costs are provided.
14. Minor issues:
Explain the decision n.1386/2013 in line28 and cite it accordingly.
Include the references of previous works missing in line 164
Correct the software name in 168: Enivimet
Correct the name of Figures : A3, A7, A1, A4, A6, A11 etc. throughout the article
Reviewer 2 Report
This paper proposed an innovative method to classify open space according to its physical characteristics and its contribution to climate vulnerability, which will have a certain influence on future studies on the regulation of urban microclimate and sustainable urban development by open spaces.
There are some problems, which must be solved before it is considered for publication. If the following problems are well-addressed, this reviewer believes that the essential contribution of this paper are important for urban sustainable development.
1. The overall framework of the article needs to be improved, and there are two titles of the fifth chapter. It is suggested to standardize the article for readers to read. For example, the article should contain the Introduction, Study area and methodology, Result, Conclusion etc.
2. Is it more appropriate to put the definition of open space in Chapter 1 rather than Chapter 2?
3. There are some spelling mistakes in Chapter 2 of the article.
For example, "Enivimet " which is in line 168, the correct spelling form is Envimet.
4. In the interpretation and calculation of each index in the second chapter of the paper, "LPT" in the index IAI is only abbreviated but not given the full name, which causes some confusion to readers.
5. Chapter 5 of the article——Classification of open spaces based on the costs and benefits and definition of the decision support tool, it is better to subdivide the title (such as the level of IPl value and the level of ICTl value), so as to make the article clearer and more concise, and not cause a certain part to be too lengthy.
The whole of the first chapter needs to be improved. The description of some sentences is not smooth and the logic between the upper and lower sentences is not strong.
Firstly, the pros and cons of planting trees may not be relevant to the overall topic of the article.
Secondly, it would be better and more complete to concentrate the definition of NBS in one paragraph.
Reviewer 3 Report
A METHOD PROPOSAL TO ADAPT URBAN OPEN BUILT AND GREEN SPACES TO CLIMATE CHANGE
Minor revision is recommended to this submission.
· Attempt seems to be good.
· Under methodology section the steps involved may be given as flowchart for ease understanding
· Geographical coordinates of the study area may be provided
· Statistical data related to microclimate changes in green and open built areas should be provided
· The legends in most of the figures are not clear especially neighbour hoods, census tracts and buildings
· The interventions to combat climate change may be discussed in detail with all the related factors
· The knowledge gap should be clearly stated.
· Care should be taken to avoid typographical error
· References should be arranged as per the journal format.
These corrections would improve the content of the manuscript and significantly increase the citations
Minor editing of English language is required
Reviewer 4 Report
Dear Editor,
I would like to thank you for your confidence in reviewing this manuscript.
I send you here my comments for the manuscript review.
Type of manuscript: Article
Title: A method proposal to adapt urban open built and green spaces to climate change.
Remarks/suggestions
Abstract:
Line 13: Please, add a coma before “which”.
Line 16: Replace “show” with “showed”.
Aim of the work:
Line 39: Put “SDGs” after its signification before us it as abbreviation.
Line 56: What does mean “WHO”?
Methodology:
Line 169: Replace “1°C” by “1 °C”.
Figures:
Figure 7: Please, change “,”with dot “.”.
Other figures and tables, change coma “,”with dot “.” And dot “.” With coma “,”.
References:
Please check references (in text and list) in relation to the journal's recommendations.
Round 2
Reviewer 1 Report
În my opinion the quality of the paper has been improved and may be published.